# Synthesis and preclinical testing of a selective beta-subtype agonist of thyroid hormone receptor ZTA-261
Masakazu Nambo [1,2,10] ✉, Taeko Nishiwaki-Ohkawa [1,3,10] ✉, Akihiro Ito[3,10], Zachary T. Ariki[1,10], Yuka Ito[3], Yuuki Kato[3], Muhammad Yar[1,7], Jacky C. -H. Yim [1], Emily Kim [3], Elizabeth Sharkey[3], Keiko Kano [1], Emi Mishiro-Sato [1], Kosuke Okimura[3], Michiyo Maruyama [3], Wataru Ota [3], Yuko Furukawa[1], Tomoya Nakayama [3], Misato Kobayashi[4,8], Fumihiko Horio[4,9], Ayato Sato [1,6] ✉, Cathleen M. Crudden [1,5] ✉ & Takashi Yoshimura [1,3,6] ✉

## Abstract

**Background** Thyroid hormones (TH) regulate the basal metabolic rate through their receptors THRα and THRβ. TH activates lipid metabolism via THRβ, however, an excess amount of TH can lead to tachycardia, bone loss, and muscle wasting through THRα. In recent years, TH analogs that selectively bind to THRβ have gained attention as new agents for treating dyslipidemia and obesity, which continue to pose major challenges to public health worldwide.

**Methods** We developed a TH analog, ZTA-261, by modifying the existing THRβ-selective agonists GC-1 and GC-24. To determine the THRβ-selectivity of ZTA-261, an in vitro radiolabeled TH displacement assay was conducted. ZTA-261 was intraperitoneally injected into a mouse model of high-fat diet-induced obesity, and its effectiveness in reducing body weight and visceral fat, and improving lipid metabolism was assessed. In addition, its toxicity in the liver, heart, and bone was evaluated.

**Results** ZTA-261 is more selective towards THRβ than GC-1. Although ZTA-261 is less effective in reducing body weight and visceral fat than GC-1, it is as effective as GC-1 in reducing the levels of serum and liver lipids. These effects are mediated by the same pathway as that of $T_3$, a natural TH, as evidenced by similar changes in the expression of TH-induced and lipid metabolism-related genes. The bone, cardiac, and hepatotoxicity of ZTA-261 are significantly lower than those of GC-1.

**Conclusions** ZTA-261, a highly selective and less toxic THRβ agonist, has the potential to be used as a drug for treating diseases related to lipid metabolism.

## Plain language summary

Nearly 10% of the world's population suffers from obesity or is overweight. These conditions are closely related to disorders of lipid metabolism, posing significant challenges to individuals and healthcare systems. Thyroid hormone (TH) activates metabolism by binding to specific protein partners, called TH receptors (THRs). There are two types of THRs, THRα and THRβ. THRβ activates lipid metabolism; however, THRα negatively affects the heart, bone, and muscle when TH is in excess. This study developed a drug called ZTA-261 that selectively binds to THRβ. Its administration to mice with induced obesity from a high-fat diet resulted in reduced body fat without any apparent toxicity. Therefore, ZTA-261 is a promising candidate to improve lipid metabolism and address the obesity epidemic.

The worldwide prevalence of obesity has doubled from 1980 to 2015, with nearly 10% of the world's population now classified as obese or overweight[1]. Obesity adversely affects the physiological functions of the body and increases the risk of developing multiple diseases, including diabetes[2], cardiovascular disease[2,3], and cancer[4]. Statins and Ezetimibe, which inhibit the biosynthesis of cholesterol in the liver and absorption of cholesterol from the small intestine, respectively, have been used as therapeutic agents for obesity but are not effective in all patients[5].

In recent years, thyroid hormones (TH) have gained attention as a new treatment for obesity[6,7]. The effects of thyroid hormones are mediated by their receptors (THRs), which belong to the nuclear receptor superfamily. Thyroxine ($T_4$), a prohormone produced in the thyroid gland, is converted to biologically active triiodothyronine ($T_3$) by iodothyronine deiodinases 1 and 2 (DIO1 and DIO2)[8]. THR forms a complex with the retinoid X receptor (RXR) and a corepressor, such as NcoR or SMRT, on the thyroid hormone-responsive elements (TRE) of target genes. The binding of $T_3$ to

A full list of affiliations appears at the end of the paper. ✉e-mail: mnambo@itbm.nagoya-u.ac.jp; tohkawa@agr.nagoya-u.ac.jp; ayato-sato@itbm.nagoya-u.ac.jp; cruddenc@chem.queensu.ca; takashiy@agr.nagoya-u.ac.jp

THR facilitates the dissociation of the corepressor and the recruitment of a coactivator, such as steroid receptor coactivator-1 (SRC-1), leading to the transcriptional activation of target genes[9]. There are two subtypes of THRs, α, and β; the α subtype (THRα) is highly expressed in the brain, heart, and muscle, while the β subtype (THRβ) is expressed mostly in the liver and in the pituitary[10].

$T_3$ has the ability to activate the metabolism of lipids in the liver and adipose tissues via THRβ, while an excess amount of $T_3$ can lead to tachycardia, as well as, bone loss and muscle wasting through THRα[11]. The biological functions induced by THRβ binding are, therefore, beneficial to address abnormal lipid metabolism and obesity, but $T_3$ shows almost no selectivity between receptors, making it difficult to avoid severe side effects derived from binding to THRα[12].

To overcome this issue, considerable efforts have been devoted to the development of THRβ-selective TH analogs by derivatizing $T_3$[13,14]. GC-1, developed by Scanlan et al., is the most representative of THRβ-selective TH analogs, showing a 10-fold lower affinity for THRα than $T_3$ without significant loss of affinity to THRβ[15]. Although the effects on lipid metabolism of GC-1 have been demonstrated in numerous in vitro and animal models, clinical trials were terminated after phase 1[13]. GC-24 is another THRβ selective agonist showing ten-fold greater selectivity for THRβ against THRα compared with GC-1. These two molecules have high structural similarity, differing only in the presence of an isopropyl group at the 3' position in the left part of GC-1 compared to a benzyl group in the same position for GC-24[16].

This study uses cross-coupling chemistry[17] to develop more potent, more selective, and less toxic GC-1 derivatives based on the structure-receptor selectivity relationship between GC-1 and GC-24. Compound ZTA-261 has the highest affinity for THRβ after screening multiple compounds with a highly congested diarylmethane structure. This study explores the potential of ZTA-261 as a THRβ-selective synthetic thyroid hormone mimic. THRβ-selectivity is assessed using a competitive ligand binding assay using full-length THRs. ZTA-261 is administered to high-fat diet-induced obesity model mice to evaluate its effect on lipid metabolism by quantifying body weight, epididymal adipose weight, and levels of total cholesterol and triglycerides in the serum and liver. The expression of lipid metabolism-related genes in the liver is analyzed to gain insight into the mechanism of action of ZTA-261. The toxicity of ZTA-261 to the liver, heart, and bone is evaluated by measuring serum alanine aminotransferase (ALT) levels and by histological analyses of the heart and bone.

## Methods
### Construction of expression vectors of THRα and THRβ
The coding sequences of full-length human *THRα* and *THRβ* were amplified by PCR. SC307938 (OriGene) and pF1KB3732 (Kazusa DNA Res. Inst.) were used as templates. The primer sequences used were as follows:

THRα sense: 5'-CGGAATTCATGGAACAGAAGCCAAGCAAGG-3'
THRα antisense: 5'-TGCGGTACCTTAGACTTCCTGATCCTC-3'
THRβ sense: 5'-CGGAATTCATGACTCCCAACAGTATGACAG-3'
THRβ antisense: 5'-GGCTCTAGATTAATCCTCGAACACTTCCA AGAAC-3'

The PCR products of *THRα* and *β* were cloned into EcoRI–KpnI and EcoRI–XbaI sites of pTNT (Promega), respectively. The resultant expression vectors, pTNT-THRα and pTNT-THRβ, were used for the synthesis of THRs by in vitro translation.

### In vitro translation of full-length THRs
THRα and THRβ were synthesized by in vitro translation using TNT T7 Quick Coupled Transcription and Translation kit (Promega) according to the manufacturer's protocol. After incubation at 30 °C for 90 min, the reaction mixtures were stored at −80 °C until use.

### 3,5,3'-[$^{125}$I]-Triiodothyronine ([$^{125}$I]-$T_3$)-displacement assay
The [$^{125}$I]-$T_3$-displacement assay was performed as described by Chapo et al.[18] with slight modifications. 2 μL of in vitro translation mixture of either

THRα or THRβ were mixed with 0.5 nM [$^{125}$I]-$T_3$ (7.4 MBq/mL, 81.4 TBq/mmol, PerkinElmer) and serially diluted non-radioactive $T_3$ or TH analogs from $10^{-5}$ to $10^{-11}$ M in 100 μL of E400 buffer (20 mM potassium phosphate buffer, pH 8.0, 400 mM NaCl, 0.5 mM EDTA, 1 mM MgCl$_2$, 1 mM monothioglycerol 500 μg/mL calf thymus histone). The reaction mixture was incubated overnight at 4 °C. A nitrocellulose membrane (Optitran, 0.45 μm in pore size, Whatman) pre-blocked by E-400 buffer was placed on a dot-blot apparatus (Minifold I, GE Healthcare), and the reaction mixtures were filtered through the membrane followed by three washes with 200 μL E-400 buffer. The membranes were removed from the manifold, allowed to air-dry for 5 min, wrapped in plastic film, and exposed to an imaging plate (BAS IP MS 2040 E, FUJI FILM) for 4 h. Radioactive signals were scanned using Typhoon FLA9000 (GE Healthcare) and quantified by Image Quant TL software (GE Healthcare). The IC$_{50}$ values for each competitor were calculated by fitting the dose–response data to the log (inhibitor) vs. response (three parameters) model using GraphPad Prism (version 8.0; GraphPad Software).

### Animals and treatment
Male 7-week-old C57BL/6J mice were obtained from a local supplier (Nihon SLC, Shizuoka, Japan). The mice were individually housed in cages (Innocage; ORIENTAL GIKEN) at 23 °C with a 12-h-light–dark cycle starting at 08:00 a.m. After one week of acclimation, the animals were randomly divided into each experimental group and fed either a normal diet (ND) (10 kcal% fat) (D12450B; RESEARCH DIETS) or a high-fat diet (HFD) (60 kcal% fat) (D12492; RESEARCH DIETS). After 8 weeks of rearing on an ND or HFD, the animals were intraperitoneally injected with either saline, $T_3$ at 0.1 or 1 μmol/kg day (65 or 650 μg/kg day), GC-1 at 0.1 or 1 μmol/kg day (33 or 330 μg/kg day), or ZTA-261 at 0.1 or 1 μmol/kg day (43 or 430 μg/kg day) for three weeks. The injection was prepared by dissolving the compounds in 0.5 N NaOH at a concentration of 100 mM, which was then further diluted with saline to a concentration of either 0.05 or 0.5 mM. The volume of the solution injected was determined based on the body weight of the animal, measured once a week, with 0.05 and 0.5 mM solutions injected at 0.1 and 1 μmol/kg day, respectively. Each group consisted of 10 animals, except for the ND ($n = 8$) and HFD with saline (vehicle, $n = 9$) groups. In the ND-treated group, one animal died during the experimental period, and one animal was excluded from the analysis after finding tumor-like lesions in the liver. In the HFD-treated group, one animal was excluded from the analysis because its body weight was less than two standard deviations below the mean.

The injection was given 6 h after light on. Food consumption and body weights were measured weekly. Food and water were provided to the animals ad libitum. The animals were treated according to the guidelines of Nagoya University, and all experimental procedures were approved by the Animal Experiment Committee of Nagoya University (A210715-003).

### Tissue collection
After three weeks of daily injections, the body weights of the mice were measured. Whole blood was collected through an abdominal aorta puncture under isoflurane anesthesia. Subsequently, the animals were euthanized using deep isoflurane anesthesia, and the liver, heart, and epididymal adipose tissues were removed from the body. Liver tissues for lipid analysis were weighed and stored at −80 °C, and for the histological experiment were frozen on dry ice embedded in Surgipath FSC 22 (Leica Biosystems) and stored at −80 °C. Hearts were weighed and fixed in 4% paraformaldehyde for 48 h in a cassette (Sakura Finetek Japan, Tokyo, Japan). Femurs were isolated, wrapped in KimWipes (Kimberly Clark), soaked in saline, and stored at −80 °C. Epididymal adipose tissue was weighed.

### Hepatic lipid analysis
Frozen livers were homogenized with chloroform-methanol (2:1), and the chloroform layer into which the liver lipids were extracted was collected. This extract was used to measure hepatic total lipid, triglyceride (TG), and total cholesterol (TC) contents. The TG and TC levels were measured using

LabAssay Cholesterol and LabAssay Triglyceride (FUJIFILM Wako Pure Chemicals), respectively, according to the manufacturer's protocol. The total liver lipid content was measured according to the method described by Forch et al.[19].

## Oil Red O staining of the liver section

20 μm sections were prepared from frozen liver tissues using Cryostat (CM3050 S, Leica). Sections were affixed to silane-coated glass slides, dried at room temperature, and stored at −80 °C until use. Before staining, sections were fixed with 1% PFA/1 × PBS, pH 7.4 (Gibco) for 5 min, immersed in 1 × PBS for 5 min, distilled water for 5 min, and 60% 2-propanol solution for 1 min. Oil Red O staining solution was prepared by mixing 0.36 g of Oil Red O (Sigma) with 120 mL of 2-propanol and 80 mL of distilled water, allowing the solution to stand at room temperature for 1 h, then filtering the solution and leaving it overnight at 37 °C. Sections were immersed in the Oil Red O staining solution at 37 °C for 15 min, rinsed in 60% 2-propanol for 1 min, and in distilled water for 1 min. Sections were counterstained with Meyer's hematoxylin. Sections were observed under a microscope (BX43; Olympus) at ×100 magnification, and digital images were captured using CellSens software (Olympus). The area of red-stained fat droplets was quantified using Image J (National Institutes of Health), and the percentage of the stained area in the total image area was calculated. Nine different parts in three consecutive sections per mouse were analyzed.

## Gene expression analysis by real-time quantitative PCR (RT-qPCR)

Total RNA was extracted from frozen liver tissue using the QIAzol reagent (Qiagen) according to the manufacturer's protocol. Extracted RNA was stored at −80 °C until use. One microgram of total RNA was subjected to cDNA synthesis using the ReverTra Ace qPCR RT Kit (TOYOBO). qPCR reaction was performed using QuantStudio 3 (Applied Biosystems) in 20 μL of reaction mixture containing 2 μL of cDNA, 10 μL of TB Green Premix Ex Taq™ II (TAKARA), 0.4 μL of ROX DyeII (TAKARA) and 0.4 μM each of gene-specific primers. *Rps18* was selected as an internal control gene from the Mouse Housekeeping Gene primer set (TAKARA) using RefFinder software[20,21] (Supplementary Fig. 1). Relative expression was calculated using Pfaffl's method[22]. The sequences of the PCR primers used were as follows:

*Cyp7a1* Forward: 5'-AGCAACTAAACAACCTGCCAGTACTA-3'
*Cyp7a1* Reverse: 5'-GTCCGGATATTCAAGGATGCA-3'
*Srebp1-c* Forward: 5'-GGAGCCATGGATTGCACATT-3'
*Srebp1-c* Reverse: 5'-AGGAAGGCTTCCAGAGAGGA-3'
*Ldlr* Forward: 5'-TGACGGGCTGGCGGTAGACTG-3'
*Ldlr* Reverse: 5'-AGTGTGATGCCATTTGGCCAC-3'
*Pnpla2* Forward: 5'-CAACCTTCGCAATCTCTAC-3'
*Pnpla2* Reverse: 5'-TTCAGTAGGCCATTCCTC-3'
*Dio1* Forward: 5'-CCCCTGGTGTTGAACTTTG A-3'
*Dio1* Reverse: 5'-CTGTGGCGTGAGCTTCTTC-3'
*Me1* Forward: 5'-AGTATCCATGACAAAGGGCAC-3'
*Me1* Reverse: 5'-ATCCCATTACAGCCAAGGTC-3'
*Thrsp* Forward: 5'-AAACCAGCGAGGCTGAGAACGA-3'
*Thrsp* Reverse: 5'-CAGGTGGGTAAGGATGTGATGG-3'

## Serum lipid and ALT analysis

The blood samples were left to stand for 1 h at room temperature, then centrifuged at 4 °C for 15 min at $800 \times g$, and the supernatant was collected. Serum TC and TG levels were measured using LabAssay Cholesterol and LabAssay Triglyceride, respectively (FUJIFILM Wako Pure Chemical Industries), according to the manufacturer's protocol. Serum ALT levels were measured using Alanine Aminotransferase (ALT or SGPT) Activity Colorimetric/Fluorometric Assay Kit (BioVision) according to the manufacturer's protocol.

## Hematoxylin–eosin staining of the heart section

Heart tissues were fixed, dehydrated, and permeated with paraffin using an automatic fixation and encapsulation device (Sakura Finetech, Japan).

Paraffin blocks were prepared using Tissue Tech TEC 6 (Sakura Finetech, Japan), and 5 μm-thick transverse transmural sections were prepared with a microtome (Leica). The sections were deparaffinized, rehydrated, and immersed in Meyer's hematoxylin solution (FUJIFILM Wako Pure Chemical Industries), followed by eosin staining. Sections were dehydrated and mounted using Entellan New (Merck). Sections were observed under a microscope (BX43; Olympus) at × 400 magnification and digital images were captured using cellSens (Olympus). 100 cells were randomly selected from the left ventricle of an individual mouse, and their trans-nuclear widths were measured according to Oláh et al.[23] using Image J (National Institute of Health).

## X-ray micro-computed tomography (μCT) of isolated mouse femur

μCT scanning of the mouse distal femur was performed using TDM 1000 (Yamato Scientific) with the following conditions: X-ray tube potential, 60 kV; tube current, 60 μA; voxel size, 9.2 μm; integration time, 125 ms; projections, 1200. A brass filter of 0.1 mm thickness was placed in the beam path to reduce the beam hardening effect. Image analysis was performed using TRI/3D-BON-FCS64 (RATOC System Engineering). Trabecular bone parameters, including bone volume fraction (BV/TV), trabecular thickness (Tb.Th), trabecular separation (Tb.Sp), trabecular number (Tb.N), and connectivity density (Conn. D) were calculated from the region 0.3–2.3 mm proximal to the epiphyseal growth plate. Five samples were randomly selected for analysis.

## Measurement of serum T$_3$ concentration

To eliminate proteins from the serum samples, 300 μL of acetonitrile containing 1 μg/mL of gabapentin as an internal standard was added to the serum. The mixture was then placed on ice for 15 min and centrifuged for 15 min at $20,000 \times g$. MonoSpin C18 cartridges (GL Science, Tokyo, Japan) were pre-conditioned with 300 μL of acetonitrile, followed by 300 μL of purified water, loaded with 200 μL of supernatants, and then washed with 300 μL of purified water. Elution was performed with 100 μL of acetonitrile. The eluate was dried, reconstituted with 30 μL of purified water, and injected into the LC–MS system.

A Cadenza CD-C18 column (3 × 150 mm, 3 μm; Imtakt, Kyoto, Japan) was used for chromatographic separation. All the experiments were performed using an Ultimate 3000 UHPLC system connected to a Q Exactive Plus Orbitrap mass spectrometer (Thermo Fisher Scientific). The injection volume for all the samples was 10 μL. The mobile phase used was 100% purified water (A) and 100% acetonitrile (B). The following gradient elution program was used: 0–2 min (5% B), 2–6 min (5–95% B), 6–8 min (95% B), 8–8.1 min (95-5% B), and 8.1–12 min (5% B) at a flow rate of 0.4 mL/min. The temperature of the autosampler and the column were maintained at 10 and 37 °C, respectively.

The MS parameters were set as follows: sheath gas flow rate, 40 psi; Aux gas flow rate, 10; spray voltage, 4 kV; capillary temperature, 350 °C; resolution, 70,000; AGC target, $3 \times 10^6$; maximum IT, 200 ms. Detection was performed in positive ion mode for gabapentin ($m/z = 172.1322$) and T$_3$ ($m/z = 651.80$). Data were acquired and processed by the Xcalibur data system (Version 4.4, Thermo Fisher Scientific). The peak areas of the analytes were normalized to those of the internal standard.

## Statistics

Data in Fig. 2 are expressed as mean ± standard deviation (SD), and subjected to non-linear regression analysis. Data in Figs. 3, 4, 7, and 8 are expressed as mean ± standard error (SEM). Data in Figs. 3, 4, 7, and 8 were evaluated using one-way ANOVA followed by Tukey's post-hoc test. Data in Figs. 5b and 6 are expressed as median, maximum, minimum, and interquartile values and were evaluated using the Kruskal–Wallis test, followed by Dunn's multiple comparisons test. All statistical tests were performed using GraphPad Prism (version 7.0, version 8.0, or version 10.0; GraphPad Software).

**Fig. 1 | Design and synthesis of THRβ-selective TH analogs.** To enhance the hydrophobic interaction with THRβ, the 3'-substituted phenol unit was replaced with naphthol. Ortho methyl groups were replaced with bulky trifluoromethyl groups, which can provide a more rigid conformation that will be effective for binding with THRs and are more electronically similar to the iodide groups in natural $T_3$ (**a**). The representative analogs (**b**), and synthetic scheme for ZTA-245 and 261 (**c**) were shown. Bn benzyl, DIBAL-H diisobutylaluminium hydride, DCM dichloromethane, TFA trifluoroacetic acid, DMF N,N-dimethylformamide, Tf tri-fluoromethanesulfonyl, EtOAc ethyl acetate.

## Reporting summary

Further information on research design is available in the Nature Portfolio Reporting Summary linked to this article.

## Results

### Design and synthesis of THRβ-selective TH analogs

We began our investigation by designing a new THRβ-selective TH analog based on key structural features of $T_3$, GC-1, and GC-24. The hydrophobic isopropyl and benzyl groups at the 3' position of the phenol in GC-1 and GC-24 have been shown to be critical for high THRβ-selectivity (Fig. 1a)[15,16]. Notably, X-ray crystallographic analysis of human THRβ with TH analogs has shown that these substituents are located in the hydrophobic region of the hormone-binding pocket, resulting in improved fit and increased affinity for THRβ[16]. The 3,5-dimethyl substituents mimic iodide substituents in the natural hormone from a space-filling point of view and, thus, are important structural elements that fix the two aromatic rings in a perpendicular arrangement. This places the molecule in the active conformation seen for effective TH analogs[24,25]. Therefore, we introduced a naphthol group to enhance the hydrophobic interaction with THRβ and increase structural rigidity. We also employed trifluoromethyl groups[26] to enhance the rigid conformation while mimicking the electronic properties of natural iodide substituents. New thyroid hormone analogs were synthesized, three of which are shown in Fig. 1b. The synthetic route to ZTA-245 and ZTA-261 is shown in Scheme (Fig. 1c). Methyl 4-bromo-2,6-bis(trifluoromethyl) benzoate **1**, which was prepared from 1,3-bis(trifluoromethyl)benzene[27], was subjected to Cu-catalyzed cross-coupling with benzyl alcohol to afford the ester **2**. The reduction of **2** followed by chlorination gave benzyl chloride **3** as an activated coupling partner. In the presence of a simple Pd catalyst,

Suzuki–Miyaura cross-coupling of **3** with (4-methoxynaphthalen-1-yl) boronic acid proceeded smoothly, giving the corresponding diarylmethane **4**. After removal of the benzyl group, the resulting phenol **5** was alkylated with t-butyl bromoacetate to afford ester **6**. Basic hydrolysis and deme-thylation provided ZTA-245. For the synthesis of ZTA-261, common intermediate **5** was converted into the triflate **7**, which underwent Pd-catalyzed Migita–Kosugi–Stille cross-coupling with allyltributylstannane to form alkene derivative **8**. Finally, oxidative cleavage of the alkene unit followed by demethylation furnished ZTA-261.

The affinity and selectivity of these new hormone derivatives for THRα and β were tentatively evaluated by a cell-based assay. In this assay, the binding of TH analogs to the ligand binding domain of either THRα or β is converted to luminescence via a luciferase reporter (Supplementary Fig. 2). From these studies, ZTA-261 became the focus as it was found to have the highest affinity for THRβ among the compounds studied (Supplementary Table 1).

### THRβ selectivity assessment of TH analogs using full-length THR

The THRβ-selectivity of $T_3$, GC-1, and ZTA-261 were assessed using a radioligand-displacement assay. Full-length human THRα and THRβ were synthesized by in vitro transcription-translation and were incubated with a fixed concentration of $[^{125}I]$ $T_3$ in the presence of non-labeled TH or TH-analogs at the concentrations indicated in Fig. 2, followed by filtration with a nitrocellulose membrane to separate the THR-bound and THR-free compounds.

Using this assay, $T_3$, GC-1, and ZTA-261 were shown to bind with similar affinity to THRβ, with $IC_{50}$ values of 3.6, 3.4, and 6.3 nM, respectively (Fig. 2 and Table 1). The key difference between these

**Fig. 2 | [$^{125}$I]-T$_3$-displacement assay.** In vitro translation mixture of either THRα or THRβ was incubated with 0.5 nM of [$^{125}$I]-T$_3$ and serially diluted non-radioactive T$_3$ (**a**), GC-1 (**b**), or ZTA-261 (**c**). [$^{125}$I]-T$_3$ bound to THRs was captured on a nitrocellulose membrane, and signals were quantified and plotted; the signals of [$^{125}$I]-T$_3$ bound to THRs in the absence of non-radioactive compounds were set to 100%. Data are shown as mean ± SD ($n = 3$). Data were fitted to the log (inhibitor) vs. response (three parameters) model using GraphPad Prism (version 8.0; GraphPad Software).

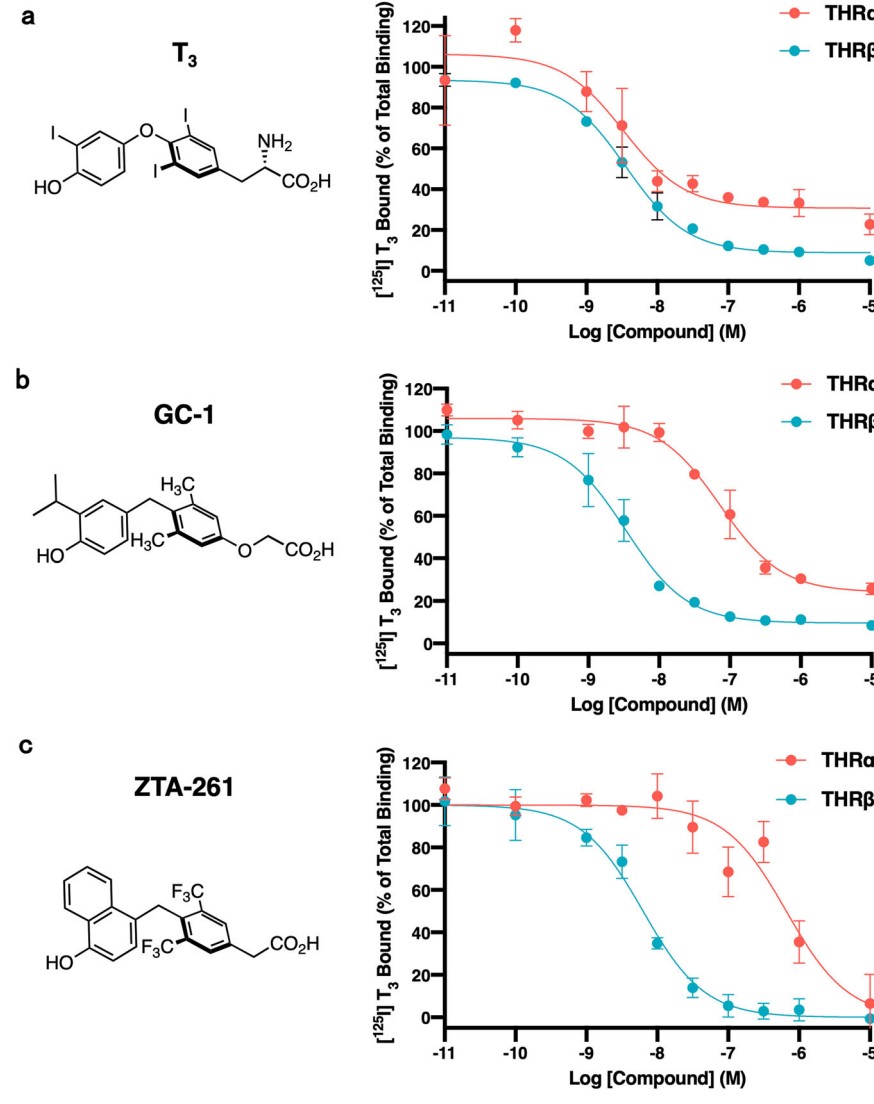

## Table 1 | IC$_{50}$ values of T$_3$, GC-1, and ZTA-261

| Competitor | THRα | | THRβ | |
|---|---|---|---|---|
| | IC$_{50}$ (nM) | 95% CI (nM) | IC$_{50}$ (nM) | 95% CI (nM) |
| T$_3$ | 3.4 | 1.8–6.5 | 3.6 | 3.0–4.3 |
| GC-1 | 73 | 53–97 | 3.4 | 2.6–4.3 |
| ZTA-261 | $6.6 \times 10^2$ | $3.4 \times 10^2$–$1.2 \times 10^3$ | 6.3 | 4.8–8.3 |

compounds is seen in their binding to THRα. While T$_3$ binds to THRα with approximately the same affinity as it binds THRβ (IC$_{50}$ of 3.4 nM), GC-1 and ZTA-261 show a much lower affinity for THRα (IC$_{50}$ of 73 and 660 nM, respectively). Thus, the affinities of GC-1 and ZTA-261 to THRα are approximately 20- and 100-fold lower than that to THRβ, respectively (Fig. 2 and Table 1). These results suggest that ZTA-261 is a highly THRβ-selective analog.

### Effects of TH analogs on body weight and accumulation of visceral fat
To assess the effect of TH analogs in vivo, T$_3$, GC-1, and ZTA-261, either at low (0.1 µmol/kg day) or high (1 µmol/kg day) doses, were injected intraperitoneally into the high-fat diet (HFD)-induced obesity model mice according to the previous report[28]. During the experimental period, food

consumption did not significantly differ between the vehicle and experimental groups (Supplementary Fig. 3).

At sacrifice, the body weight of HFD-fed mice injected with the vehicle was ~1.5 times higher than that in ND-treated mice (Fig. 3a). Mice injected with GC-1 at both low and high doses and high doses of T$_3$ and ZTA-261 had decreased body weight by ~20% compared to those treated with vehicle (Fig. 3a). Treatment with low doses of T$_3$ and ZTA-261 did not significantly reduce body weight (Fig. 3a), suggesting that ZTA-261 can reduce body weight in mice to the same extent as the natural thyroid hormone T$_3$, but less effectively than GC-1. The weight of the epididymal adipose tissue decreased significantly following T$_3$, GC-1, and a high dose of ZTA-261 treatment, whereas a low dose of ZTA-261 did not show any obvious effect (Fig. 3b).

### Effects of TH analogs on lipid metabolism
To investigate the effects of the TH analogs on lipid metabolism, we measured the levels of total cholesterol and triglycerides in plasma and liver, and total lipids in liver. Serum cholesterol levels decreased in all experimental groups compared to the vehicle control group (Fig. 4a). Serum triglyceride levels were also decreased by injection of GC-1 at a low dose and ZTA-261 at a high dose, whereas there was no significant difference between the T$_3$- and vehicle-injected groups (Fig. 4b). Although liver cholesterol levels showed no significant differences in any of the experimental groups (Fig. 4c), the levels of total lipids and triglycerides in the liver were significantly decreased

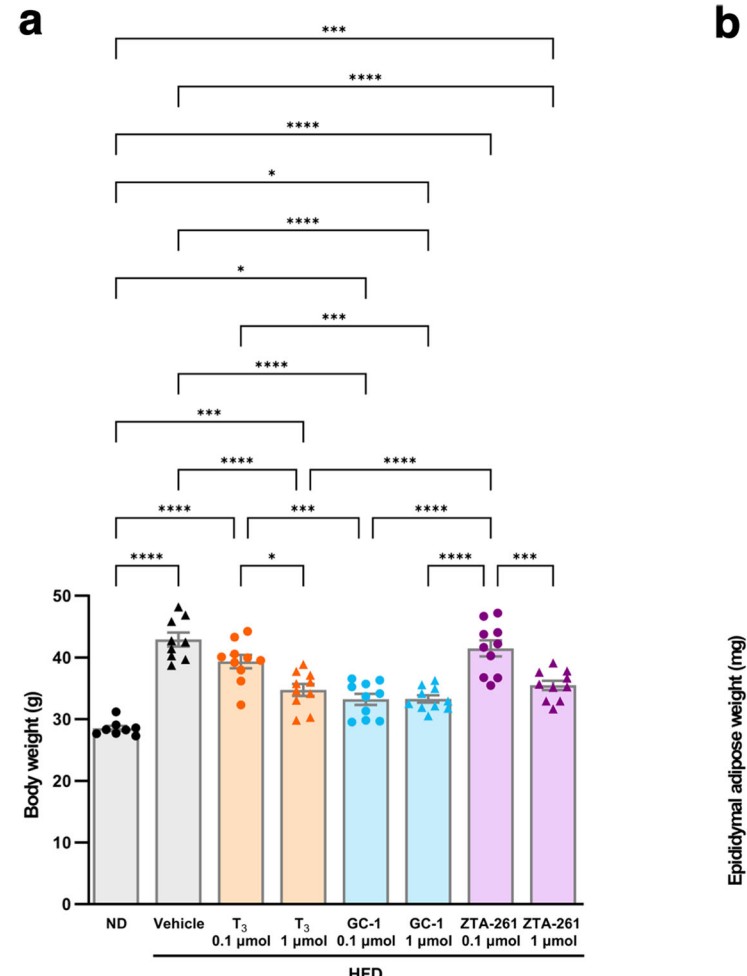

**Fig. 3 | Effects of intraperitoneal injection of TH analogs on body weight and deposition of visceral fat.** Male 8-weeks-old C57BL/6J mice were fed either a normal diet (ND; 10 kcal% fat) or a high-fat diet (HFD; 60 kcal% fat) and maintained for 8 weeks. Then, animals were intraperitoneally injected with either saline (Vehicle), $T_3$ (0.1 or 1 μmol/kg day), GC-1 (0.1 or 1 μmol/kg day), or ZTA-261 (0.1 or 1 μmol/kg day). After three weeks of injection, body weight (**a**), and weight of the epididymal adipose tissue (**b**) were measured. Data are shown as the mean ± SEM ($n = 8$–10). Significant differences among groups are indicated by asterisks (*$P < 0.05$, **$P < 0.01$, ***$P < 0.001$, and ****$P < 0.0001$, one-way ANOVA with Tukey's post-hoc test).

by GC-1, ZTA-261, and high-dose $T_3$ treatment (Fig. 4d, e). These results suggest that ZTA-261 can improve lipid metabolism in mice as effectively as GC-1.

## Histological evaluation of lipid accumulation in the liver

Excessive accumulation of lipid droplets in hepatocytes is characteristic of non-alcoholic fatty liver disease (NAFLD)[29]. To evaluate the effect of the TH analogs on liver lipids, liver sections were stained with Oil Red O to visualize the accumulation of lipid droplets in hepatocytes (Fig. 5a and Supplementary Fig. 4). Treatment with ZTA-261 at a dose of 1 μmol/kg·day significantly reduced the Oil Red O-positive area compared to the vehicle to a level as low as the ND-treated group (Fig. 5a, b).

## Gene expression analysis in the liver

To explore the mechanism by which ZTA-261 promotes lipid metabolism, we examined the expression of THR-regulated genes and genes involved in lipid metabolism in the liver[30] by reverse transcription-quantitative PCR (RT-qPCR) analysis upon administration of the compound at 1 μmol/kg day (Fig. 6). We first examined the expression of THR-regulated genes, including *Iodothyronine deiodinase 1* (*Dio1*), *Malic enzyme 1* (*Me1*), and *Thyroid hormone responsive* (*Thrsp*). Dio1 catalyzes the conversion of $T_4$ to $T_3$. *Me1* encodes an enzyme that generates NADPH for fatty acid synthesis. Thrsp is involved in triglyceride

biosynthesis. The expression of *Dio1* and *Me1* were significantly upregulated by $T_3$, GC-1, and ZTA-261 (Fig. 6a, b). GC-1 and ZTA-261 increased the expression of *Thrsp*, but $T_3$ showed no significant difference compared with the vehicle (Fig. 6c).

We checked the expression levels of four lipid metabolism-related genes, *Cytochrome P450, family 7, subfamily a, polypeptide 1* (*Cyp7a1*), *Low-density lipoprotein receptor* (*Ldlr*), *Sterol regulatory element binding protein-1c* (*Srebp1c*), and *Patatin-like phoshodomain containing 2* (*Pnpla2*). CYP7A1 is a rate-limiting enzyme involved in cholesterol catabolism. LDLR promotes the cellular uptake of LDL and facilitates cholesterol degradation. The expression of *Cyp7a1* (Fig. 6d) and *Ldlr* (Fig. 6e) in the $T_3$-, GC-1-, and ZTA-261-treated groups was not significantly different from that in the vehicle-treated group. The expression level of *Srebp1c*, which encodes a transcription factor that activates the synthesis of fatty acids and triglycerides, was significantly lower in the $T_3$ and GC-1 groups than in the vehicle group. Although not statistically significant, the ZTA-261-treated group showed decreased *Srebp1c* expression (Fig. 6f). Levels of *patatin-like phosphodomain-containing 2* (*Pnpla2*), also known as *adipose triglyceride lipase* (*ATGL*), which encodes an enzyme that catalyzes the hydrolysis of triglycerides[31], were significantly higher in the GC-1 and ZTA-261-treated group than in the vehicle group (Fig. 6g). These results suggest that GC-1 and ZTA-261 promote lipid metabolism by activating similar pathways.

**Fig. 4 | Effects of TH analogs on the accumulation of lipids in mice.** Mice fed either a normal diet (ND; 10 kcal% fat) or a high-fat diet (HFD; 60 kcal% fat) were intraperitoneally injected with either saline, $T_3$ (0.1 or 1 µmol/ kg·day), GC-1 (0.1 or 1 µmol/kg·day) or ZTA-261 (0.1 or 1 µmol/kg·day). After 3 weeks of injection, serum and liver tissues were collected, and the amount of serum cholesterol (**a**), serum triglycerides (**b**), liver cholesterol (**c**), liver triglycerides (**d**), and liver total lipids (**e**) were measured. Data are shown as the mean ± SEM ($n = 8$–10). Significant differences among groups are indicated by asterisks (*$P < 0.05$, **$P < 0.01$, ***$P < 0.001$, and ****$P < 0.0001$, one-way ANOVA with Tukey's post-hoc test).

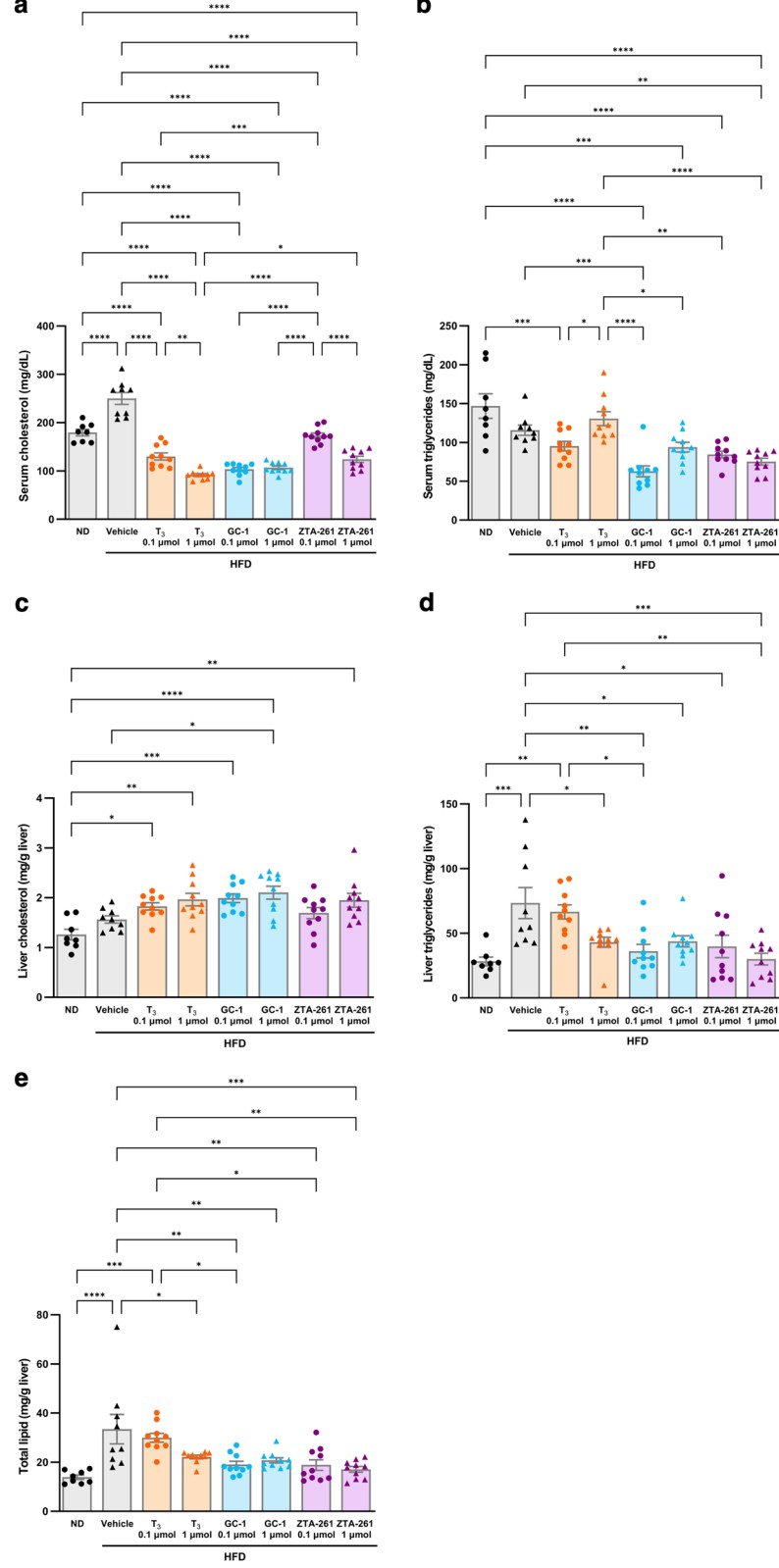

## Toxicity assessment of TH analogs in the liver and heart

To evaluate the hepatotoxicity of ZTA-261 in vivo, we measured the serum levels of alanine aminotransferase (ALT), a marker of hepatocyte necrosis. A significant increase in serum ALT level was observed when GC-1 was administered at 1 µmol/kg day compared to vehicles (Fig. 7a). On the other hand, ZTA-261-administered mice did not show a significant increase in

serum ALT levels compared to vehicles at either 0.1 or 1 µmol/kg day (Fig. 7a), suggesting that the ZTA-261 is less hepatotoxic than GC-1.

Left ventricular hypertrophy is often associated with hyperthyroidism caused by increased heart rate mediated by human THRα[32]. Therefore, THRβ-selective analogs are expected to circumvent this adverse effect on the heart. We measured the heart weight of mice treated with the $T_3$ or TH

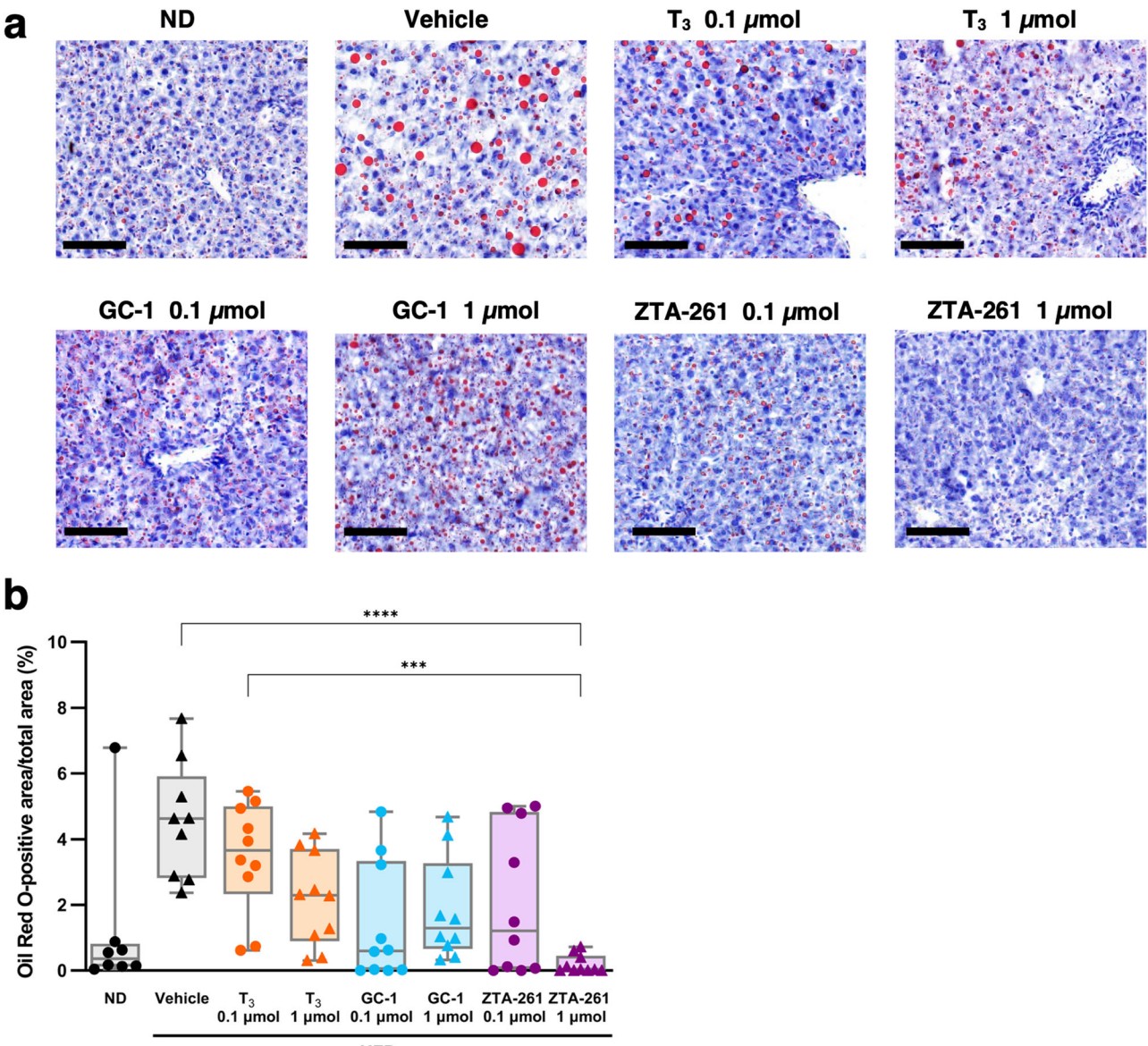

**Fig. 5 | Oil Red O staining of the liver section of mice treated with TH analogs.** Mice fed either a normal diet (ND; 10 kcal% fat) or a high-fat diet (HFD; 60 kcal% fat) were intraperitoneally injected with either saline, $T_3$ (0.1 or 1 μmol/kg day), GC-1 (0.1 or 1 μmol/kg day) or ZTA-261 (0.1 or 1 μmol/kg day). After three weeks of injection, liver tissues were collected, frozen sections were prepared, and subjected to Oil Red O staining (**a**). Stained areas were quantified and normalized to the total image area. Data are shown as box plots, with median, 25th and 75th percentiles and maximum and minimum values ($n = 8$–10). Significant differences among groups are indicated by asterisks (***$P < 0.001$ and ****$P < 0.0001$, Kruskal–Wallis test with Dunn's multiple comparisons test) (**b**).

analogs to examine whether cardiac hypertrophy was induced. Administration of both high (1 μmol/kg day) and low (0.1 μmol/kg day) doses of $T_3$ or a high dose of GC-1 caused a significant increase in heart weight compared to vehicles (Fig. 7b). No such effect was found in the heart of animals treated at either high or low doses of ZTA-261 (Fig. 7b). To test whether the increase in heart weight resulted from an increase in the size of the cardiomyocytes in the left ventricle, we stained the heart sections with hematoxylin-eosin and measured the transnuclear width of the left ventricular cardiomyocytes (Supplementary Fig. 5 and Fig. 7c). Both high and low doses of $T_3$, a high dose of GC-1, and a high dose of ZTA-261 administration significantly increased the width of the myocytes. However, the effect sizes were larger in animals treated with $T_3$ compared to those treated with GC-1 or ZTA-261 (Fig. 7c).

**Toxicity assessment of TH analogs in bone**
Bone morphometry was performed using X-ray micro-computed tomography (μCT) to evaluate the side effects of the TH analogs on bone. The metaphysis of the distal femur was analyzed (Fig. 8a), and the following parameters describing the microstructure of the trabecular bone were measured: BV/TV (%), Tb. N (mm$^{-1}$), Tb. Sp (μm), Tb. N (μm) and Conn. D (mm$^{-3}$) (Fig. 8b–f). BV/TV indicates the fraction of bone volume (BV) in the total volume of interest (TV). The BV/TV value in the 1 μmol $T_3$-treated group was significantly lower than those in the vehicle- and 1 μmol ZTA-261-treated groups (Fig. 8b). A significant decrease in BV/TV was also observed in the 1 μmol GC-1-treated group compared to the vehicle group, but not between the vehicle and ZTA-261-treated groups (Fig. 8b). Tb. N is the number of trabeculae per unit length. Tb. N was the lowest in the 1 μmol $T_3$-treated group, followed by the 1 μmol GC-1-treated group. The Tb. N values were significantly lower in these two groups than those in the vehicle- and ZTA-261-treated groups (Fig. 8c). Tb. Sp indicates the mean distance between trabeculae. It was significantly higher in the 1 μmol $T_3$-treated group than that in the other groups. The ZTA-261-treated group did not show a significant difference compared with the

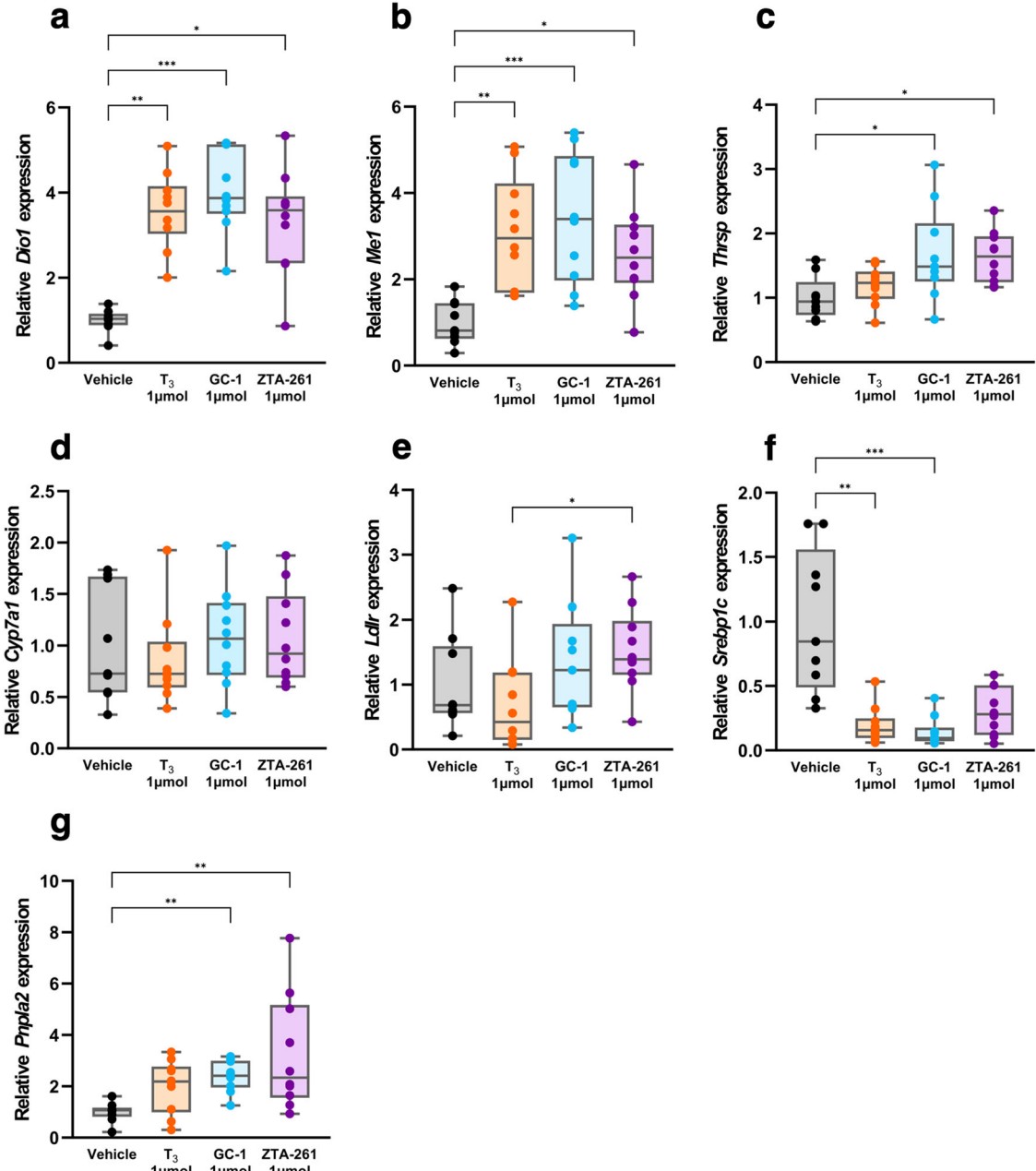

**Fig. 6 | Effect of TH analogs on the expression of THR-regulated genes and lipid metabolism-related genes in the liver.** Total RNA was extracted from frozen liver tissue from mice fed with a high-fat diet administered either saline, T₃, GC-1, or ZTA-261 at 1 μmol/kg day, and subjected to RT-qPCR analysis. Relative expression levels of *Dio1* (**a**), *Me1* (**b**), *Thrsp* (**c**), *Cyp7a1* (**d**), *Ldlr* (**e**), *Srebp1c* (**f**), and *Pnpla2* (**g**) were determined by Pfaffl's method using *Rsp18* as a reference gene. Data are shown as box plots, with median, 25th and 75th percentiles and maximum and minimum values ($n = 9$–10). Significant differences among the groups are indicated by different letters Significant differences among groups are indicated by asterisks (*$P < 0.05$, **$P < 0.01$, and ***$P < 0.001$, Kruskal–Wallis test with Dunn's multiple comparisons test).

vehicle-treated group (Fig. 8d). There was no significant difference in Tb. Th, trabecular thickness, in the four groups (Fig. 8e). Conn. D. characterizes the redundancy of trabecular connections normalized by TV. It was significantly decreased in the T₃ and GC-1-treated groups compared to that in the vehicle groups (Fig. 8f). A slight decrease was also observed in the ZTA-261 treated group compared to that in the vehicle group, but the decrease was not significant (Fig. 8f). These parameters except Tb. Th were severely affected in the T₃-treated group, which is consistent with the fact that T₃ has no selectivity for THRα and β. From these results, the harmful effects on the bone were reduced in the THRβ-selective analogs; however, when comparing GC-1 and ZTA-261, the toxicity was further decreased in ZTA-261.

## Blood–brain barrier permeability of TH analogs

TH stimulates the differentiation of oligodendrocyte progenitors and myelin production during central nervous system (CNS) development. In addition to its developmental functions, TH facilitates remyelination after pathological demyelination in the adult CNS[33]. Thus, TH and its analogs are potential treatments for CNS disorders such as multiple sclerosis.

Previously, the CNS distribution of GC-1 was assessed in mice and was found to be distributed to the CNS with efficiency at the lowest limit for compounds approved as CNS drugs[34]. Permeability of the blood-brain barrier (BBB) is a key property of drugs that determines their distribution to the CNS. Therefore, we tested BBB transparency in vitro using the BBB kit RBT-24H (PharmaCo-Cell)[35]. We applied 10 μM of either GC-1 or ZTA-

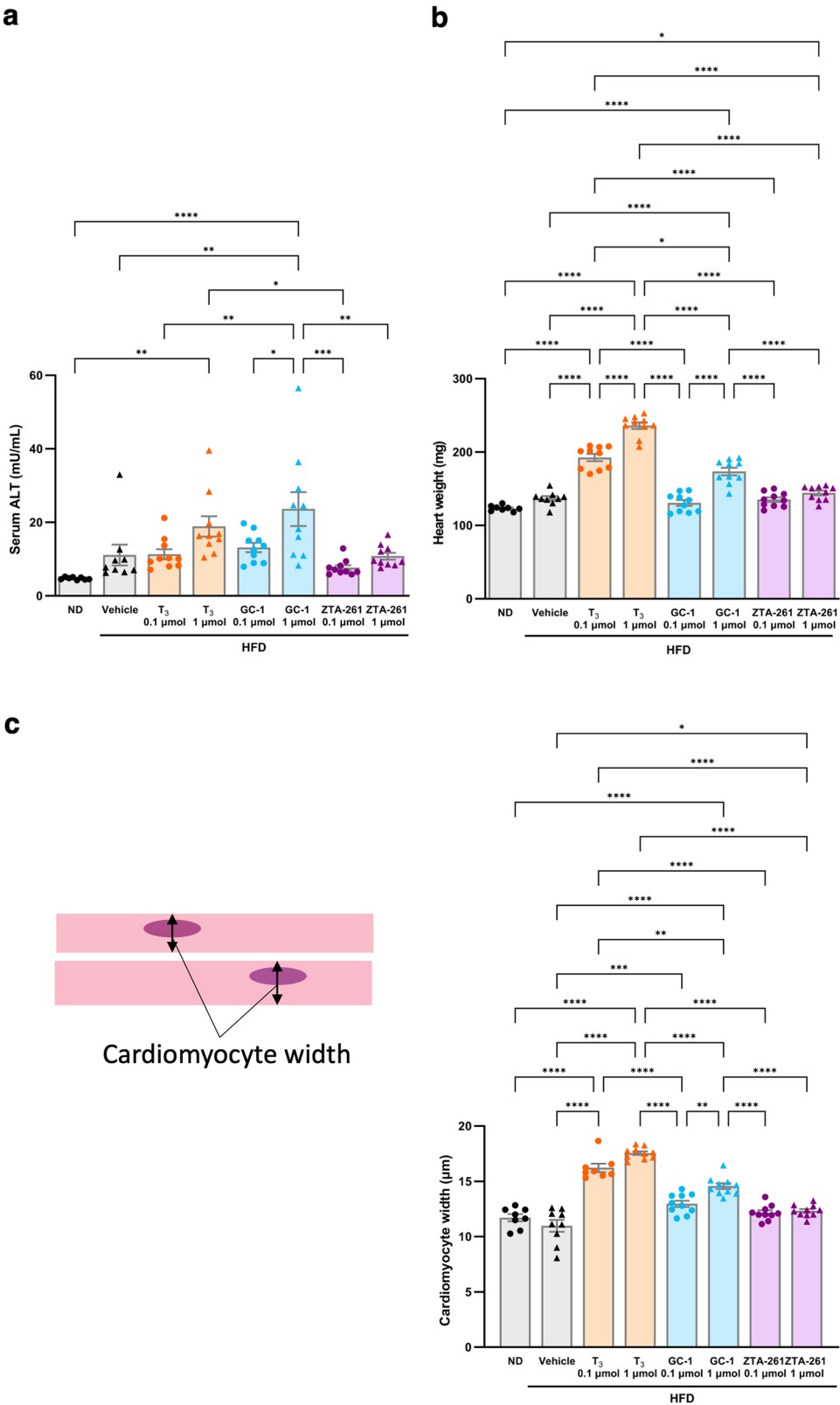

**Fig. 7 | Toxicity assessment of TH analogs on the liver and heart.** Serum was collected from mice fed with either a normal diet or a high-fat diet injected with either saline, $T_3$ (0.1 or 1 μmol/kg day), GC-1 (0.1 or 1 μmol/kg day), or ZTA-261 (0.1 or 1 μmol/kg day) and ALT levels were measured to assess hepatic toxicity of compounds. Data are shown as the mean ± SEM ($n = 8$–10). Significant differences among groups are indicated by asterisks (*$P < 0.05$, **$P < 0.01$, ***$P < 0.001$, and ****$P < 0.0001$, one-way ANOVA with Tukey's post-hoc test) (**a**). Heart weight was measured to assess heart toxicity. Data are shown as the mean ± SEM

($n = 8$–10). Significant differences among groups are indicated by asterisks (*$P < 0.05$, and ****$P < 0.0001$, one-way ANOVA with Tukey's post-hoc test) (**b**). Heart sections were prepared and subjected to hematoxylin–eosin staining. 100 cells were randomly selected from the left ventricle of an individual mouse, their trans-nuclear widths were measured, and average values were calculated. Data are shown as the mean ± SEM ($n = 8$–10). Significant differences among groups are indicated by asterisks (*$P < 0.05$, **$P < 0.01$, ***$P < 0.001$, and ****$P < 0.0001$, one-way ANOVA with Tukey's post-hoc test) (**c**).

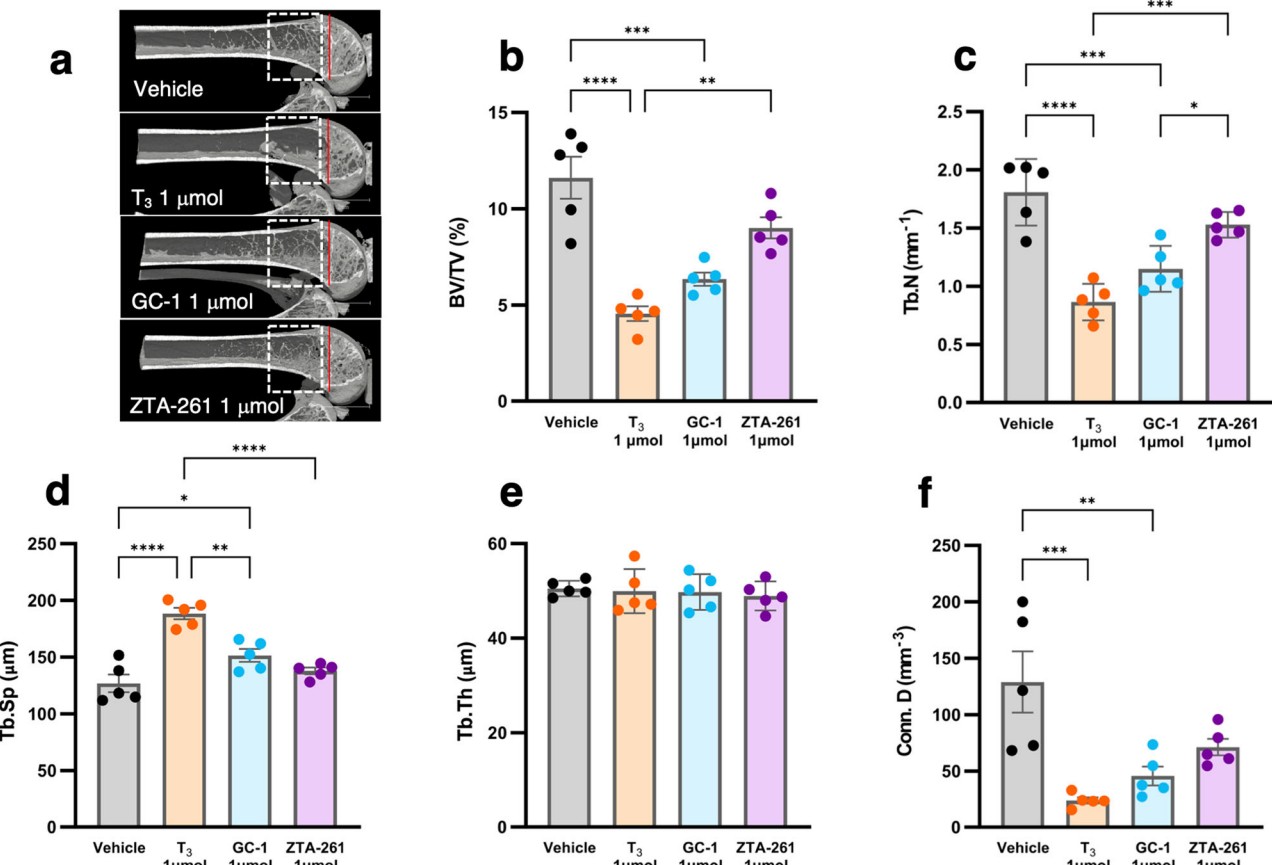

**Fig. 8 | Toxicity assessment of TH analogs on the bone by μCT analysis.** Femur was collected from mice fed with a high-fat diet (60 kcal% fat) injected with either saline, $T_3$ (1 μmol/kg day), GC-1 (1 μmol/kg day), or ZTA-261 (1 μmol/kg day). The trabecular region of the distal femur of each mouse was subjected to μCT scanning. Representative μCT images of the distal femur (**a**). The analyzed region, indicated by a white rectangle, spanned 0.3–2.3 mm proximal to the red line marking the epiphyseal growth plate. Trabecular bone parameters including bone volume fraction (BV/TV) (**b**), trabecular number (Tb.N) (**c**), trabecular separation (Tb.Sp) (**d**), trabecular thickness (Tb.Th) (**e**), and connectivity density (Conn.D) (**f**) were calculated. Data are shown as the mean ± SEM ($n = 5$). Significant differences among groups are indicated by asterisks (*$P < 0.05$, **$P < 0.01$, ***$P < 0.001$, and ****$P < 0.0001$, one-way ANOVA with Tukey's post-hoc test).

261 to the wells, mimicking the luminal side of the BBB, and allowed them to penetrate into the wells mimicking the abluminal side for 30 min. Caffeine and digoxin were used as the positive and negative controls, respectively. The apparent permeabilities ($10^6$ cm/s) of these compounds were calculated using the following equation:

$$P_{\text{app}} = \frac{V_A}{A \cdot [C]_{luminal}} \frac{\Delta[C]_{Abluminal}}{\Delta T} \qquad (1)$$

where $V_A$ is the volume on the abluminal side, $A$ is the surface area, $[C]_{luminal}$ is the initial concentration of drugs on the luminal side, $\Delta[C]_{Abluminal}$ is the change in the concentration of drugs on the abluminal side within the assay period $\Delta T$ (30 min). $P_{app}$ values of GC-1 and ZTA-261 were 14.6 ± 3.54 and 7.28 ± 2.55, respectively, while caffeine, known as a CNS stimulant, showed $P_{app}$ values of 76.4 ± 8.35 (Supplementary Table 2). The $P_{app}$ value of digoxin could not be determined because the concentration of digoxin on the abluminal side was below the detection limit (0.1 μM). These results indicate that GC-1 had limited permeability to the CNS, which is consistent with a previous report[34], and that the permeability of ZTA-261 was even smaller than that of GC−1. These results suggest that ZTA-261 is particularly useful for treating liver hyperlipidemia without affecting the CNS. Future studies are required to identify transporters for GC-1 and ZTA-261.

### Effects of TH analogs on serum $T_3$ levels
To investigate the impact of the TH analogs on endogenous TH levels, we measured the concentration of $T_3$ in serum samples collected from animals injected with either $T_3$, GC-1 or ZTA-261 at 1 μmol/kg day (Table 2). In the $T_3$-injected animals, the serum $T_3$ concentration was 743.5 ± 78.0 ng/dL, which was about 10 times higher than the control group, while in the GC-1 and ZTA-261-injected groups, the amount of $T_3$ is below the quantification limit. The increase in $T_3$ levels in the $T_3$-injected group may reflect exogenous $T_3$. It is possible that GC-1 and ZTA-261 inhibit endogenous $T_3$ synthesis and secretion by acting on the pituitary gland, which does not have a BBB, but on the hypothalamus.

### Discussion
Several THRβ-selective agonists have been developed recently[36–38], and one such molecule, MGL-3196, is currently under phase 3 trials for the treatment of non-alcoholic steatohepatitis (NASH) and NAFLD[39]. Considering this, THRβ is a promising drug target for improving lipid metabolism and NASH/NAFLD.

In vitro THR competition assays revealed that ZTA-261 binds to THRβ with a high affinity, comparable to $T_3$ and GC-1, and with higher THRβ selectivity than GC-1 (Fig. 1, Table 1). The effect of ZTA-261 on lipid metabolism was tested in vivo using a mouse model of high-fat diet-induced obesity (Fig. 4). The results showed that administration of both GC-1 and ZTA-261 led to a significant reduction in the levels of liver total cholesterol, triglycerides, and serum triglycerides. This finding is consistent with the fact that the $IC_{50}$ values of GC-1 and ZTA-261 for THRβ were quite similar.

We examined the expression of genes regulated by THR and those related to lipid metabolism in the liver using RT-qPCR (Fig. 6). Consistent with previous studies[40,41], the administration of $T_3$, GC-1, and ZTA-261

**Table 2 | Serum T$_3$ concentration**

|  | Vehicle | T$_3$ 1 μmol/ kg day | GC-1 1 μmol/kg day | ZTA-261 1 μmol/kg day |
|---|---|---|---|---|
| Serum T$_3$ (ng/dL) | 85.4 ± 16.0 | 743.5 ± 78.0 | B.L.Q. | B.L.Q. |

Data were shown as mean ± SEM (*n* = 9–10). One data point of 1 μmol/kg day of GC-1 was excluded from the analysis due to a lack of internal control value.
B.L.Q.; below the limit of quantification.

significantly increased *Dio1* and *Me1* expression. GC-1 and ZTA-261 also upregulated *Thrsp*, whereas T$_3$ tended to increase *Thrsp*, although not significantly. These results suggest that these compounds act via THR-dependent signaling (Fig. 6a–c).

Previous studies have shown that GC-1 reduces serum cholesterol levels by inducing *Cyp7a1*[42,43]. However, another study showed that these compounds have dose-dependent effects on expression levels of *Cyp7a1* in mice[28]. The greatest increase in *Cyp7a1* was observed at a dose of 20 nmol/kg/day for both T$_3$ and GC-1, with a gradual decrease as the dose increased[28]. In this study, *Cyp7a1* expression was not affected by T$_3$, GC-1, or ZTA-261 (Fig. 6d and e), which may be explained by the high doses (i.e., 0.1 and 1 mmol/kg/day).

TH is believed to reduce serum cholesterol levels by stimulating *Ldlr*[44]. However, experiments using *Ldlr*$^{-/-}$ mice have shown that TH can decrease serum cholesterol levels independently of Ldlr[28,42,43,45]. Furthermore, GC-1 treatment did not always induce *Ldlr* expression[42]. These findings indicate that treatment with TH and its analogs does not necessarily increase *Ldlr* expression.

It is worth noting that the results of our gene expression analysis were consistent with the observation that ZTA-261, T$_3$, and GC-1 were more effective at reducing triglyceride levels than cholesterol levels in the liver (Fig. 4c, d). Treatment with T$_3$ and GC-1 significantly decreased the expression of *Srebp1c*, activating the synthesis of triglycerides, and treatment with ZTA-261 significantly upregulated the expression of *Pnpla2*, regulating the degradation of triglycerides (Fig. 6f, g). On the other hand, expression of *Cyp7a1* and *Ldlr* was not affected by these compounds (Fig. 6d, e). These results explain why the administration of these compounds decreased triglyceride levels more efficiently than cholesterol.

ZTA-261 was more effective at reducing fat droplets in the liver. The administration of ZTA-261 at 1 μmol/kg·day reduced fat droplets to a level comparable to that in animals fed a normal diet. In contrast, the liver TG levels were similar in the GC-1- and ZTA-261-treated groups. This discrepancy may be explained by the fact that the number of lipid droplets is also increased by drug-induced liver injury[46]. In this study, serum ALT levels were higher in the GC-1-treated groups than in the ZTA-261-treated groups. The droplets observed in the GC-1-treated group could be partly explained by the side effects of GC-1 on the liver.

T$_3$ causes harmful effects on the bone and cardiac system by binding to THRα, which is a major barrier to its use as an obesity treatment[47,48]. When we examined serum T$_3$ levels, neither ZTA-261 nor GC-1 induced hyperthyroidism (Table S2). We observed an increase in heart weight in the T$_3$- and 1 μmol GC-1-treated groups but not in ZTA-261-treated groups at any concentration (Fig. 7b). We conducted a micro-CT analysis to evaluate the impact of T$_3$, GC-1, and ZTA-261 on bone structure (Fig. 8). The results showed that administration of T$_3$ led to significant damage to the trabecular bone structure of the distal femur as indicated by the BV/TV, Tb.N., Tb.Sp., and Conn.D. values. This harmful effect was mitigated in the GC-1-treated group. The ZTA-261 treatment group did not show any significant differences in these values compared with the control groups. The differences in bone and cardiac toxicity among T$_3$, GC-1, and ZTA-261 can be attributed to the degree of THRβ selectivity. T$_3$, being non-selective, had a more harmful impact on the heart and bone than THRβ-selective GC-1 and ZTA-261. Comparatively, ZTA-261 was less toxic than GC-1, possibly due to its higher selectivity for THRβ than GC-1. Since white fat is considered to have more THRα activity than THRβ activity, the reduction in epididymal fat by GC-1 may also reflect the degree of THRβ selectivity.

Drug-induced liver injury is known to be one of the major factors in the discontinuation of clinical trials[49]. The hepatotoxicity of ZTA-261 was evaluated by quantifying alanine aminotransferase (ALT) released in the blood. Administration of T$_3$ and GC-1 at a dose of 1 μmol/kg significantly increased serum ALT levels, while ZTA-261, when administered at the same concentration, did not. This indicated that ZTA-261 did not cause hepatotoxicity at the dosage required to promote lipid metabolism (Fig. 7a). Notably, the variability in the ALT levels in individual animals was lower in the ZTA-261-treated groups than in the GC-1-treated groups (Fig. 7a). From a clinical perspective, small individual differences in the harmful effects of a compound are considered an appropriate property as a lead for further modifications. Further pharmacokinetics and pharmacodynamics studies are required to establish the potential of ZTA-261 as a pharmaceutical treatment. In this study, we administered a higher dose of compounds for a longer duration than usual to investigate potential side effects caused by THRα. In future studies, the dosage and duration should be carefully considered to further evaluate the effects of ZTA-261.

In conclusion, we developed a TH analog ZTA-261 and assessed its effects and toxicity in vitro and in vivo. ZTA-261 has high affinity and selectivity for THRβ, which accelerates liver lipid metabolism with low toxicity in the heart and bone. These results suggested that ZTA-261 could potentially be used as a drug lead for diseases related to lipid metabolism, such as NAFLD.

## Data availability
All source data for the figures and tables presented in this article can be found in the Supplementary Data 1 to 8 and Supplementary Information files. All other data are available from the corresponding authors upon reasonable request.

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

## Acknowledgements

We would like to thank Hiroaki Oda, Atsushi Murai, Eiichi Hondo, and Atsuo Iida for their helpful discussions and advice, RATOC System Engineering Co., Ltd. (Tokyo, Japan) for μCT imaging and analysis of the mouse femur, and Editage (www.editage.jp) for English language editing. This work was supported by JSPS KAKENHI, Grant Number JP19H03178 (T.N.-O.), JP24H00058 (T.Y.), JP19H05643 (T.Y.), and JP26000013 (T.Y.). M.N. thanks the Nakatomi Foundation, the Pharmacological Research Foundation, and the Ono Medical Research Foundation for their support. We acknowledge the JSPS and NU for funding this research through the World Premier International Research Center Initiative (WPI) Program and Center for One Medicine Innovative Translational Research (COMIT).

## Author contributions

M.N., T.N.-O., A.S., C.M.C., and T.Y. designed the study. M.N., T.N.-O., Z.T.A., A.I., Y.I., Y.K., M.Y., J.C.-H.Y., E.K., E.S., K.K., E.M.-S., K.O., M.M., W.O., Y.F., T.N., A.S., and T.Y. performed the experiments. M.K., F.H., C.M.C., and T.Y. contributed new reagents/analytic tools. M.N., T.N.-O., Z.T.A., A.I., Y.I., Y.K., M.Y., and J.C.-H.Y. analyzed the data, and M.N., T.N.-O., A.I., A.S., C.M.C., and T.Y. wrote the manuscript.

## Competing interest

The authors declare no competing interests.

## Additional information

[1]Institute of Transformative Bio-Molecules (WPI-ITbM), Nagoya University, Furo-cho, Chikusa-ku, Nagoya 464-8601, Japan. [2]Department of Chemistry, Graduate School of Science, Nagoya University, Furo-cho, Chikusa-ku, Nagoya 464-8601, Japan. [3]Laboratory of Animal Integrative Physiology, Department of Animal Sciences, Graduate School of Bioagricultural Sciences, Nagoya University, Furo-cho, Chikusa-ku, Nagoya 464-8601, Japan. [4]Laboratory of Animal Nutrition, Department of Animal Sciences, Graduate School of Bioagricultural Sciences, Nagoya University, Furo-cho, Chikusa-ku, Nagoya 464-8601, Japan. [5]Department of Chemistry, Queen's University, Chernoff Hall, Kingston, ON K7L 3N6, Canada. [6]Center for One Medicine Innovative Translational Research (COMIT), Nagoya University, Nagoya 464-8601, Japan. [7]Present address: Interdisciplinary Research Center in Biomedical Materials, COMSATS, University Islamabad Lahore Campus, Lahore 54000, Pakistan. [8]Present address: Department of Nutritional Sciences, Nagoya University of Arts and Sciences, Nisshin, Aichi 470-0196, Japan. [9]Present address: Department of Life Studies and Environmental Science, Nagoya Women's University, Nagoya 467-8610, Japan. [10]These authors contributed equally: Masakazu Nambo, Taeko Nishiwaki-Ohkawa, Akihiro Ito, Zachary T. Ariki. ✉e-mail: mnambo@itbm.nagoya-u.ac.jp; tohkawa@agr.nagoya-u.ac.jp; ayato-sato@itbm.nagoya-u.ac.jp; cruddenc@chem.queensu.ca; takashiy@agr.nagoya-u.ac.jp

