## [Peer Review File · Communications Medicine]

Reviewers' comments:

Reviewer #1 (Remarks to the Author):

This work by Nishiwaki-Ohkawa explores the synthesis of new T3 analogs and their effect on fat accumulation in mice. Others have well explored this field. However, their new compound appears to have better selectivity for THRB than THRA. Unfortunately, the data are limited in studying the major problem with these analogs: non-specific effects. For example, we are not told if the animals are systemically thyrotoxic, which would produce a general decrease in fat accumulation. Also, are the animals tachycardia? Does the compound penetrate the BBB? Previous analogs have used the first-pass effect to more specifically target the liver, realizing that the analogs aren't that organ-specific. These are just some of the questions that have been raised about the use of T3 analogs and their safety.

Reviewer #2 (Remarks to the Author):

The manuscript by Nambo et.al focuses on the discovery of a novel Thyroid hormone analog ZTA-261 that is able to bind the THRB receptor selectively and with low toxicity. The major claim of this study is that ZTA-261 is a drug to ameliorate lipid metabolism with low toxicity effects. The data presented are convincing and combining biochemical studies and in vitro and in vivo data provide a new molecule that acting specifically on THRB could improve lipid metabolism.

Major revision

1) To explore the mechanism by which ZTA-261 promotes lipid metabolism the authors performed RT-qPCR analysis for four genes involved in lipid metabolism.

Honestly, it would be more convincing to perform RNA-seq in mice treated with ZTA-61, GC-1, T3 and vehicles fed with high-fat diet to dissect the molecular pathway.

Minor point

1) Fig 2A. please change in the word "Compond" with "Compound"

Reviewer #3 (Remarks to the Author):

See attachment.

Reviewer #4 (Remarks to the Author):

The manuscript entitled „Development of a novel thyroid hormone analog ZTA-261 selective for β -subtype of thyroid hormone receptor with low toxicity” by Nambo et al studies a new TRb drug in

comparison to the established GC-1 in a mouse model. The in vitro data suggest a TRb preference of the new compound that is at least comparable to GC-1. The in vivo data demonstrate weight loss on HFD comparable to T3 but less pronounced than GC1, and beneficial effects on liver lipid metabolism.

Specific concerns:

- 1) The in vivo dose used for all drugs is massive and in the high pharmacological range. This is evidenced by the observed weight loss of the mice – usually mice in contrast to rats gain weight upon T3 treatment. It is therefore questionable whether this condition reflects the range that would be used for therapeutic approaches. I agree that the dose is useful to identify possible side-effects of the drug, but the study falls short in this regard, as only cardiomyocyte width and heart weight have been recorded – especially since cardiac hypertrophy requires a central action of thyroid hormone. The authors need to discuss the choice of dose in greater detail and need to include additional studies on other possible TH target organs, including heart and bone, by providing qPCRs of TH target genes.
- 2) The qPCR data in Fig 6 should include additional T3 target genes such as Dio1 or Me1. Moreover, the use of GAPDH as only housekeeping gene is suboptimal. The authors should test three different housekeeping genes and find the most stable gene/combination for normalization, eg using Normfinder.
- 3) The statistical analysis using one-way ANOVA e.g. in Fig 7 is not correct, as it ignores that two different doses have been used.

Reviewer #5 (Remarks to the Author):

The manuscript deals with the design and synthesis of a new agonist of THRs, endowed of selectivity against THR β . The new compound synthesized was evaluated as potential therapeutic agent for hyperlipidemia and obesity.

The new molecule showed to reduce hepatic lipid droplets in HFD induced obesity model mice. The authors assessed also the cardio and hepatotoxicity comparing the results with GC-1, a THRBeta selective agonist which terminated Phase I clinical trials because of cardiotoxicity.

The manuscript is in a hot topic field and numerous researches are currently focused on the search for new THR beta agonists to treat NASH, NAFLD and HCC. Although it deserves publication few revisions are needed before to accept it.

1. Title should be changed, and written in a more catchy way. Just to suggest one example: Design and synthesis of ZTA-261: a selective beta-subtype agonist of thyroid hormone receptor with low toxicity
2. Abstract: I suggest to write a concise and clear abstract that resumes the results of the study. In such way, the abstract is mainly a brief introduction on the THRs' biological activity.
3. Add in the abstract also the analysis of gene expression
4. References should be updated. Several paper have been recently published describing novel THRBeta agonists (Li, Qiu, et al. Discovery of a highly selective and H435r-sensitive thyroid hormone receptor β agonist. *Journal of Medicinal Chemistry*, 2022, 65.10: 7193-7211; *Eur J Med Chem*. 2020 Feb 15;188:112006. doi: 10.1016/j.ejmech.2019.112006.) More importantly on January 2023 Resmetirom has been approved by FDA as the first THRBeta agonist to treat NASH. The authors must include these references and update the introduction with current state-of-the-art in this field. To assess whether the effect on lipid metabolism is mediated by activation of THRs, a qRT-PCR analysis of THR-target genes, such as Dio1, Thrsp, and Me1 should be performed (ref.:

<https://doi.org/10.3390/ijms222313105>)

5. Please check ref 20. The citation seems to be wrong. The article cited describes the use of fluorinated sulfone in cross-coupling reaction, but the authors cite this paper to justify the replacement of methyl groups of GC-1 with CF₃ moiety.

6. Synthetic procedures: Please add in Reagent and condition (figure 1c) the reaction yields. In the supporting info the purity of the final compounds should be assessed by HPLC or elemental analysis and data should be added in the experimental part.

7. What is the difference between table 1 and table S1 (IC₅₀ values of compounds)? Please explain and correct, appropriately

8. In the discussion section the authors justify the results hypothesizing an improved bioavailability of ZTA-261 when compared with GC-1, and an improved distribution in the liver compared with plasma. These speculations should be confirmed by experimental data. I suggest to discuss only results emerging from this study and clearly declare that further study should be performed in order to attribute these results to a different PK profile.

Comments

Nambo M. et al. developed a new T3 analog ZTA-261. Authors showed that selectivity of ZTA-261 for THR β over THR α is greater, compared to sobetirome (GC-1). Authors also showed the lipid lowering properties of ZTA-261 is comparable to GC-1. This could be useful for treatment of hypothyroidism especially to patient with heart problems, and as well as to lowering lipid without inducing cardiac thyrotoxicosis.

Specific comments

1. In this manuscript, authors did not test the drug in treatment of hypothyroid animals. Hypothyroidism increase cholesterol and TG level, which commonly seen in patients with hypothyroidism. ZTA-261 treatment of hypothyroid animals could generate helpful information about the drug.
2. Authors focused on lipid lowering using mice fed with high fat diet and showed comparable results to GC-1 and better than T3. It is known that T3 treatment induces hyperthyroidism and GC-1 (462nmol/kg/day) does not significantly alter serum and tissue T3 level (Grover et al., Endocrinology 2004). Whether ZTA-261 affects serum T3 was not investigated in the manuscript. In order to confirm ZTA-261 possess basic therapeutic properties, testing serum TH level is relevant.
3. T3 and GC-1 stimulate cholesterol catabolism and increase LDLR and CYP7a1, resulting in cholesterol efflux from liver. Liver cholesterol is expected to decrease. Authors showed (Figure 4C), T3 and analogs increased hepatic cholesterol. Please explain.
4. In this manuscript, the statistical analysis for animal experiment (different drug and concentration) requires Two-way ANOVA, not One-way ANOVA. Please revise the statistical analysis

Minor comments

Introduction

Line 64-67, “The binding of T3 to THR... such as a peroxisomal proliferator-activated receptor (PPAR) α and γ ...” The description is incorrect. PPARs are not a transcription coactivator. PPARs, same as THRs, are ligand-activated transcription factors. Please revise it.

Line 69, “...while the β subtype (THR β) is expressed mostly in the liver.”

Please note that it is well known that THR β is the dominant form of THR in pituitary as well.

Line 70-71, “When T3 binds to THR α , adverse effects such as heart arrhythmia and loss of muscle and bone mass occur”. This description is incorrect. In the normal condition of these tissues, binding THR α to T3 is required for tissue normal development and normal function. Hyperthyroidism, not T3 binds to THR α , causes tachycardia, bone loss and muscle wasting.

Line 86-88, "...to develop more potent, selective and less toxic derivatives of GC-1." This statement is premature. Authors has no solid evidence shown in the manuscript ZEA-261 is less toxic than GC-1.

Materials and Methods

1. *Animal and treatment*

Describe how the ZTA-261 drug is prepared for injection and injection volume as well. Authors used equal molar dose in T3 and analgoues injection. For easy comparison with published data, authors may also provide the injection dose in ug/kg/day.

2. 125I-T3 Ligand displacement assay

In the experiment, authors described that the ligand-THR mixture were incubated overnight at 4C. The receptor ligand bind is a quick reaction. Longer incubation increases non-specific binding. Please explain why overnight incubation is necessary.

3. Gene expression analysis by qPCR

It is now well recognized that Gapdh gene expression is influenced by T3. When studying T3/analgoues-mediated gene expression, Grapdh should be avoided if possible.

Generally, gene expression by qPTR requires multiple housekeeping genes (minimum two) to be used in normalization.

Since cholesterol lowering is not as good as GC-1 and T3, authors may consider examine the gene expression of SREBP2, SR-B1

Results

1. T3 displacement assays

- What is the molecular weight of the ZTA-261?
- Was a blank *in vitro* translation (without plasmid) included in the binding assay?
- How is non-specific binding determined? In Figure 2, nonspecific binding can be observed. However, the curve fitting omitted it.

2. For Animal experiments

- Statistical analysis for these animal experiments requires 2-way ANOVA in order to compare difference among T3, GC-1 and ZTA-261.
- After 3-weeks of ZTA-261 injection, whatwas the serum TH level?

3. Figure 7C Cardiomyocyte width. An image would be helpful.

4. Toxicity assessment

- Line 219-222, please indicate the treatment duration.

Responses to Reviewer 1

We are grateful to Reviewer 1 for the critical comments and useful suggestions that helped us improve our manuscript. As indicated in the following responses, we have considered all the comments and suggestions in the revised version of our manuscript.

REMARKS TO THE AUTHOR

This work by Nishiwaki-Ohkawa explores the synthesis of new T₃ analogs and their effect on fat accumulation in mice. Others have well explored this field. However, their new compound appears to have better selectivity for THR β than THRA. Unfortunately, the data are limited in studying the major problem with these analogs: non-specific effects. For example, we are not told if the animals are systemically thyrotoxic, which would produce a general decrease in fat accumulation. Also, are the animals tachycardia? Does the compound penetrate the BBB? Previous analogs have used the first-pass effect to more specifically target the liver, realizing that the analogs aren't that organ-specific. These are just some of the questions that have been raised about the use of T₃ analogs and their safety.

RESPONSE

Tachycardia and bone loss are major symptoms of thyrotoxicosis. As shown in Figure 7B and C, we measured heart weight and myocardial width to ascertain the compound's adverse effects on the heart; these values were elevated in the T₃ group compared to the control group, whereas no significant changes were observed in the ZTA-261 group. These results suggest that ZTA-261, unlike T₃, has little adverse effect on the heart. In this revised version, we have further added trabecular bone morphometry data in Figure 8. The results showed that T₃ caused decreased bone mass, while no significant difference was observed between the ZTA-261 and control groups. These results indicate that ZTA-261 has little negative impact on the heart and bone, due to its high selectivity for the THR β .

We have also conducted an in vitro permeability assay of the blood-brain barrier, and the results are presented in Table 2. Our data indicates that the permeability of ZTA-261 is lower than that of existing GC-1, and it does not readily reach the central nervous system.

Unlike previously reported prodrugs (such as VK2809), as mentioned above, which utilize the first-pass effect, the compounds we reported here do not contain a structural motif in their structures that would be expected to have organ-specific metabolism.

Responses to Reviewer 2

We are grateful to Reviewer 2 for the critical comments and useful suggestions that helped us improve our manuscript. As indicated in the following responses, we have considered all the comments and suggestions in the revised version of our manuscript.

REMARKS TO THE AUTHOR

The manuscript by Nambo et.al focuses on the discovery of a novel Thyroid hormone analog ZTA-261 that is able to bind the THRβ receptor selectively and with low toxicity. The major claim of this study is that ZTA-261 is a drug to ameliorate lipid metabolism with low toxicity effects. The data presented are convincing and combining biochemical studies and in vitro and in vivo data provide a new molecule that acting specifically on THRβ could improve lipid metabolism.

MAJOR REVISION

1) To explore the mechanism by which ZTA-261 promotes lipid metabolism the authors performed RT-qPCR analysis for four genes involved in lipid metabolism.

Honestly, it would be more convincing to perform RNA-seq in mice treated with ZTA-61, GC-1, T3 and vehicles fed with high-fat diet to dissect the molecular pathway.

RESPONSE

We conducted further RT-qPCR analysis on three genes regulated by THR: *Dio1*, *Me1*, and *Thrsp*. The expression of these genes was upregulated by ZTA-261, as well as by T3 and GC-1. These findings suggest that ZTA-261 regulates lipid metabolism through the thyroid hormone signaling pathway.

MINOR POINT

1) Fig 2A. please change in the word “Compond” with “Compound”

RESPONSE

We have corrected the typographical error.

Responses to Reviewer 3

We are grateful to Reviewer 3 for the critical comments and useful suggestions that helped us improve our manuscript. As indicated in the following responses, we have considered all the comments and suggestions in the revised version of our manuscript.

COMMENTS

Nambo M. et al. developed a new T3 analog ZTA-261. Authors showed that selectivity of ZTA-261 for THR β over THR α is greater, compared to sobetirome (GC-1). Authors also showed the lipid lowering properties of ZTA-261 is comparable to GC-1. This could be useful for treatment of hypothyroidism especially to patient with heart problems, and as well as to lowering lipid without inducing cardiac thyrotoxicosis.

1. In this manuscript, authors did not test the drug in treatment of hypothyroid animals. Hypothyroidism increase cholesterol and TG level, which commonly seen in patients with hypothyroidism. ZTA-261 treatment of hypothyroid animals could generate helpful information about the drug.

RESPONSE

Unfortunately, we do not have hypothyroid mice available at the moment, which means the study suggested by reviewer 3 will have to be postponed for the future. However, we were able to meet the request of reviewer 3 by measuring the amount of T₃ in the serum in each experimental group's mice. These results could be useful in discussing the potential impacts of the compounds on hypothyroidism. Please find the detailed information below.

COMMENTS

2. Authors focused on lipid lowering using mice fed with high fat diet and showed comparable results to GC-1 and better than T3. It is known that T3 treatment induces hyperthyroidism and GC-1 (462nmol/kg/day) does not significantly alter serum and tissue T3 level (Grover et al., Endocrinology 2004). Whether ZTA-261 affects serum T3 was not investigated in the manuscript. In order to confirm ZTA-261 possess basic therapeutic properties, testing serum TH level is relevant.

RESPONSE

We measured the serum T₃ level in vehicle-, T₃ (1 μmol/kg/day)-, GC-1 (1 μmol/kg/day)-, and ZTA-261 (1 μmol/kg/day)-treated mice (Table S3). In the T₃-treated group, the concentration of T₃ was elevated to 743.5 ± 78.0 (ng/dL), indicating hyperthyroidism. In contrast, ZTA-261 did not induce hyperthyroidism as in the case of GC-1

COMMENTS

3. T₃ and GC-1 stimulate cholesterol catabolism and increase LDLR and CYP7a1, resulting in cholesterol efflux from liver. Liver cholesterol is expected to decrease. Authors showed (Figure 4C), T₃ and analogs increased hepatic cholesterol. Please explain.

RESPONSE

As depicted in Figures 6D and E, Mice treated with T₃, GC-1, and ZTA-261 did not exhibit a significant increase in *Cyp7a1* and *Ldlr*, which are associated with cholesterol catabolism. However, a decrease in *Srebp1C*, which activates TG synthesis, and an increase in *Pnpla2*, which catalyzes TG degradation, were observed. We are not sure why the expression of *Cyp7a1* and *Ldlr* did not increase, but our results match our lipid level measurements (Figure 4); these analogs are more effective in decreasing the TG than cholesterol.

COMMENTS

4. In this manuscript, the statistical analysis for animal experiment (different drug and concentration) requires Two-way ANOVA, not One-way ANOVA. Please revise the statistical analysis.

RESPONSE

The main objective of this study is to compare the impact of injecting T₃, GC-1, and ZTA-261 at low and high concentrations with that of injecting vehicles. We employed one-way ANOVA to achieve this goal. Examples of similar situations can be found in the following papers on the development of THR agonists (*Front. Endocrinol.* 14, 1109615, 2023; *J. Med. Chem.* 66, 3284-3300, 2023). In addition, we conducted two-way ANOVA for the groups that were treated with either of the three compounds at two concentrations. Please note that the vehicle-treated

group cannot be included in this analysis. The results are presented below.

	0.1 $\mu\text{mol T}_3$	1 $\mu\text{mol T}_3$	0.1 $\mu\text{mol GC-1}$	1 $\mu\text{mol GC-1}$	0.1 $\mu\text{mol ZTA-261}$	1 $\mu\text{mol ZTA-261}$	Effect of compound	Effect of Dose	Interaction
Body weight (g)	39.3 \pm 1.1	34.7 \pm 1.0	33.2 \pm 0.9	33.3 \pm 0.6	41.5 \pm 1.3	35.5 \pm 0.8	$p < 0.0001$	$p < 0.0001$	$p = 0.0068$
Epididymal adipose weight (mg)	1762.6 \pm 148.0	1015.0 \pm 36.4	1485.7 \pm 103.3	1093.5 \pm 50.3	2649.6 \pm 146.3	1679.3 \pm 109.0	$p < 0.0001$	$p < 0.0001$	$p = 0.0323$
Liver triglyceride (mg/g liver)	66.54 \pm 5.46	43.04 \pm 3.90	36.11 \pm 5.30	43.75 \pm 4.29	39.81 \pm 8.66	30.01 \pm 4.48	$p = 0.0022$	$p = 0.0657$	$p = 0.0258$
Liver cholesterol (mg/g liver)	1.83 \pm 0.07	1.96 \pm 0.13	1.99 \pm 0.09	2.10 \pm 0.13	1.69 \pm 0.11	1.95 \pm 0.14	$p = 0.1375$	$p = 0.0749$	$p = 0.7966$
Liver total lipid (mg/g liver)	30.0 \pm 1.8	22.1 \pm 0.7	19.1 \pm 1.3	20.7 \pm 1.1	18.8 \pm 2.1	17.1 \pm 1.2	$p < 0.0001$	$p = 0.0303$	$p = 0.0067$
Serum triglyceride (mg/dL)	95.26 \pm 6.13	130.5 \pm 8.92	62.68 \pm 7.05	93.92 \pm 6.17	84.35 \pm 4.25	75.25 \pm 4.44	$p < 0.0001$	$p = 0.0005$	$p = 0.0014$
Serum cholesterol (mg/dL)	129.83 \pm 7.30	91.58 \pm 2.82	103.23 \pm 3.91	106.61 \pm 3.48	172.98 \pm 5.35	124.09 \pm 6.40	$p < 0.0001$	$p < 0.0001$	$p < 0.0001$
Oil Red O-positive area (%)	3.46 \pm 0.54	2.17 \pm 0.44	1.40 \pm 0.57	1.85 \pm 0.49	2.06 \pm 0.70	0.19 \pm 0.09	$p = 0.0050$	$p = 0.0337$	$p = 0.0668$
Serum ALT (mU/mL)	11.35 \pm 1.36	18.90 \pm 2.78	13.15 \pm 1.28	23.62 \pm 4.61	7.67 \pm 0.68	10.80 \pm 0.93	$p = 0.0012$	$p = 0.0006$	$p = 0.3065$
Heart weight (mg)	192.6 \pm 4.9	236.1 \pm 4.5	130.6 \pm 3.9	173.5 \pm 5.1	135.1 \pm 3.1	144.1 \pm 3.1	$p < 0.0001$	$p < 0.0001$	$p < 0.0001$
Cardiomyocyte width (μm)	16.2 \pm 0.3	17.5 \pm 0.2	13.0 \pm 0.3	14.6 \pm 0.3	12.2 \pm 0.2	12.3 \pm 0.2	$p < 0.0001$	$p < 0.0001$	$p = 0.0127$

Data are shown as mean \pm SEM (n=8-10)

MINOR COMMENTS

Introduction

Line 64-67, “The binding of T3 to THR... such as a peroxisomal proliferator-activated receptor (PPAR) α and γ ...” The description is incorrect. PPARs are not a transcription coactivator. PPARs, same as THRs, are ligand-activated transcription factors. Please revise it.

RESPONSE

As an example of a coactivator, we used SRC-1 instead of PPARs.

Line 69, “...while the β subtype (THR β) is expressed mostly in the liver.”

Please note that it is well known that THR β is the dominant form of THR in pituitary as well.

RESPONSE

We have included this point in the text.

Line 70-71, “When T3 binds to THR α , adverse effects such as heart arrhythmia and loss of muscle and bone mass occur”. This description is incorrect. In the normal condition of these tissues, binding THR α to T3 is required for tissue normal development and normal function. Hyperthyroidism, not T3 binds to THR α , causes tachycardia, bone loss and muscle wasting.

RESPONSE

We have made the following changes to the relevant sentences.

T₃ has the ability to activate the metabolism of lipids in the liver and adipose tissues via THR β , while an excess amount of T₃ can lead to tachycardia, as well as bone loss and muscle wasting through THR α (Line 71-73 in the revised version).

Line 86-88, "...to develop more potent, selective and less toxic derivatives of GC-1." This statement is premature. Authors has no solid evidence shown in the manuscript ZEA-261 is less toxic than GC-1.

RESPONSE

We performed trabecular bone morphometry (Figure 8) and found that ZTA-261 is less toxic on bone than GC-1. T₃ had severe adverse effect on bone mass, as indicated by BV/TV, Tb.N, Tb.Sp, and Conn.D values, while the ZTA-261-treated group showed no significant differences in these parameters compared to the control group. In the GC-1-treated group, these parameters showed significant differences compared to the control groups. These results suggest that ZTA-261 is less harmful to bone, possibly due to its higher selectivity for THR β than GC-1.

Materials and Methods

1. Animal and treatment

Describe how the ZTA-261 drug is prepared for injection and injection volume as well. Authors used equal molar dose in T3 and analogs injection. For easy comparison with published data, authors may also provide the injection dose in ug/kg/day.

RESPONSE

As per the reviewer's suggestion, we have included the details on the injection of compounds in the Materials and Methods section (Line 414-421 in the revised version).

2. 125I-T3 Ligand displacement assay

In the experiment, authors described that the ligand-THR mixture were incubated overnight at 4C. The receptor ligand bind is a quick reaction. Longer incubation increases non-specific binding. Please explain why overnight incubation is necessary.

RESPONSE

We conducted the assay as described in Chapo et al. (J. Pharmacol. Toxicol. Methods 56:28-33, 2007), where the reaction mixtures were incubated overnight.

3. Gene expression analysis by qPCR

It is now well recognized that Gapdh gene expression is influenced by T3. When studying T3/analogues-mediated gene expression, Gapdh should be avoided if possible. Generally, gene expression by qPCR requires multiple housekeeping genes (minimum two) to be used in normalization. Since cholesterol lowering is not as good as GC-1 and T3, authors may consider examine the gene expression of SREBP2, SR-B1

RESPONSE

After evaluating 14 different primer sets of housekeeping genes, including 13 from the Mouse housekeeping gene primer set by TAKARA and one previously used GAPDH primer set, we selected Rps18 using RefFinder software (Fig. S4).

The treatment of mice with T₃, GC-1, and ZTA-261 had no significant impact on the expression of *Cyp7a1* and *Ldlr*, which regulate cholesterol metabolism. Instead, there was a decrease in *Srebp1c*, which activates TG synthesis, and an increase in *Pnpla2*, which catalyzes TG degradation (Figure 6). These results are consistent with the observation that GC-1 and ZTA-261 are more effective in reducing TG levels than cholesterol levels (Figure 4).

Results

1. T3 displacement assays

- What is the molecular weight of the ZTA-261?

RESPONSE

The molecular weight of ZTA-261 is 428.33

- Was a blank in vitro translation (without plasmid) included in the binding assay?

RESPONSE

We did not include the in vitro translation mixture without plasmid in the blank wells.

- How is non-specific binding determined? In Figure 2, nonspecific binding can be observed. However, the curve fitting omitted it.

RESPONSE

We reanalyzed the data using GraphPad Prism, fitting with log(inhibitor) vs. response (three parameters) instead of sigmoidal dose-response. This resulted in better approximations of the top and bottom values.

2. For Animal experiments

- Statistical analysis for these animal experiments requires 2-way ANOVA in order to compare difference among T₃, GC-1 and ZTA-261.

RESPONSE

The main objective of this study is to compare the impact of injecting T₃, GC-1, and ZTA-261 at low and high concentrations with that of injecting vehicles. We employed one-way ANOVA to achieve this goal. Examples of similar situations can be found in the following papers on the development of THR agonists (*Front. Endocrinol.* 14, 1109615, 2023; *J. Med. Chem.* 66, 3284-3300, 2023). In addition, we conducted two-way ANOVA for the groups that were treated with either of the three compounds at two concentrations. Please note that the vehicle-treated group cannot be included in this analysis. The results are presented below.

	0.1 μmol T ₃	1 μmol T ₃	0.1 μmol GC-1	1 μmol GC-1	0.1 μmol ZTA-261	1 μmol ZTA-261	Effect of compound	Effect of Dose	Interaction
Body weight (g)	39.3 ± 1.1	34.7 ± 1.0	33.2 ± 0.9	33.3 ± 0.6	41.5 ± 1.3	35.5 ± 0.8	p < 0.0001	p < 0.0001	p = 0.0068
Epididymal adipose weight (mg)	1762.6 ± 148.0	1015.0 ± 36.4	1485.7 ± 103.3	1093.5 ± 50.3	2649.6 ± 146.3	1679.3 ± 109.0	p < 0.0001	p < 0.0001	p = 0.0323
Liver triglyceride (mg/g liver)	66.54 ± 5.46	43.04 ± 3.90	36.11 ± 5.30	43.75 ± 4.29	39.81 ± 8.66	30.01 ± 4.48	p = 0.0022	p = 0.0657	p = 0.0258
Liver cholesterol (mg/g liver)	1.83 ± 0.07	1.96 ± 0.13	1.99 ± 0.09	2.10 ± 0.13	1.69 ± 0.11	1.95 ± 0.14	p = 0.1375	p = 0.0749	p = 0.7966
Liver total lipid (mg/g liver)	30.0 ± 1.8	22.1 ± 0.7	19.1 ± 1.3	20.7 ± 1.1	18.8 ± 2.1	17.1 ± 1.2	p < 0.0001	p = 0.0303	p = 0.0067
Serum triglyceride (mg/dL)	95.26 ± 6.13	130.5 ± 8.92	62.68 ± 7.05	93.92 ± 6.17	84.35 ± 4.25	75.25 ± 4.44	p < 0.0001	p = 0.0005	p = 0.0014
Serum cholesterol (mg/dL)	129.83 ± 7.30	91.58 ± 2.82	103.23 ± 3.91	106.61 ± 3.48	172.98 ± 5.35	124.09 ± 6.40	p < 0.0001	p < 0.0001	p < 0.0001
Oil Red O-positive area (%)	3.46 ± 0.54	2.17 ± 0.44	1.40 ± 0.57	1.85 ± 0.49	2.06 ± 0.70	0.19 ± 0.09	p = 0.0050	p = 0.0337	p = 0.0668
Serum ALT (mU/mL)	11.35 ± 1.36	18.90 ± 2.78	13.15 ± 1.28	23.62 ± 4.61	7.67 ± 0.68	10.80 ± 0.93	p = 0.0012	p = 0.0006	p = 0.3065
Heart weight (mg)	192.6 ± 4.9	236.1 ± 4.5	130.6 ± 3.9	173.5 ± 5.1	135.1 ± 3.1	144.1 ± 3.1	p < 0.0001	p < 0.0001	p < 0.0001
Cardiomyocyte width (μm)	16.2 ± 0.3	17.5 ± 0.2	13.0 ± 0.3	14.6 ± 0.3	12.2 ± 0.2	12.3 ± 0.2	p < 0.0001	p < 0.0001	p = 0.0127

Data are shown as mean ± SEM (n=8-10)

- After 3-weeks of ZTA-261 injection, what was the serum TH level?

RESPONSE

T₃ peaks were detected in mice serum after 3 weeks of injection with either ZTA-261 or GC-1, but were below the quantification limit.

3. Figure 7C Cardiomyocyte width. An image would be helpful.

RESPONSE

The cardiomyocyte width is measured as the short axis length at the mid-nuclear level of individual cardiomyocytes. We included a diagram to illustrate this measurement in Fig. 7.

4. Toxicity assessment

- Line 219-222, please indicate the treatment duration.

RESPONSE

We collected samples for the toxicity assay from the same mice used in the other assays, with a three-week administration period.

Responses to Reviewer 4

We are grateful to Reviewer 4 for the critical comments and useful suggestions that helped us improve our manuscript. As indicated in the following responses, we have considered all the comments and suggestions in the revised version of our manuscript.

REMARKS TO THE AUTHOR

The manuscript entitled „Development of a novel thyroid hormone analog ZTA-261 selective for β -subtype of thyroid hormone receptor with low toxicity” by Nambo et al studies a new TR β drug in comparison to the established GC-1 in a mouse model. The in vitro data suggest a TR β preference of the new compound that is at least comparable to GC-1. The in vivo data demonstrate weight loss on HFD comparable to T3 but less pronounced than GC1, and beneficial effects on liver lipid metabolism.

SPECIFIC COMMENTS

1) The in vivo dose used for all drugs is massive and in the high pharmacological range. This is evidenced by the observed weight loss of the mice – usually mice in contrast to rats gain weight upon T3 treatment. It is therefore questionable whether this condition reflects the range that would be used for therapeutic approaches. I agree that the dose is useful to identify possible side-effects of the drug, but the study falls short in this regard, as only cardiomyocyte width and heart weight have been recorded – especially since cardiac hypertrophy requires a central action of thyroid hormone. The authors need to discuss the choice of dose in greater detail and need

to include additional studies on other possible TH target organs, including heart and bone, by providing qPCRs of TH target genes.

RESPONSE

We determined the dosage of drugs based on a paper by Johansson et al. (PNAS 102, 10297-10302, 2005). The study demonstrated that the administration of T3 and GC-1 at 97 nmol/kg/day was efficient in reducing serum and liver cholesterol and triglyceride levels in mice. Therefore, we decided on a dose of 0.1 μ mol/kg/day. In addition to this dosage, we selected 1 μ mol/kg/day to study potential side effects. We were unable to conduct qPCR analysis on the heart and bone tissues as they were not stored in a way that would allow for RNA preparation. Instead, we carried out a micro-CT analysis of the femur and observed that ZTA-261 had a less pronounced effect on bone architecture as compared to T3 and GC-1.

2) The qPCR data in Fig 6 should include additional T3 target genes such as Dio1 or Me1. Moreover, the use of GAPDH as only housekeeping gene is suboptimal. The authors should test three different housekeeping genes and find the most stable gene/combination for normalization, eg using Normfinder.

RESPONSE

We investigated the effects of TH analogs on three T3 target genes: Dio1, Me1, and Thrsp, and observed a significant increase in their expression. To identify a suitable housekeeping gene, we evaluated 14 different primer sets, including 13 from the Mouse housekeeping gene primer set by TAKARA, as well as a previously used GAPDH primer set. Using RefFinder software, we selected Rps18 as the most appropriate housekeeping gene (Fig.S4).

3) The statistical analysis using one-way ANOVA e.g. in Fig 7 is not correct, as it ignores that two different doses have been used.

RESPONSE

The main objective of this study is to compare the impact of injecting T₃, GC-1, and ZTA-261 at low and high concentrations with that of injecting vehicles. We employed one-way ANOVA to achieve this goal. Examples of similar situations can be found in the following papers on the

development of THR agonists (*Front. Endocrinol.*14, 1109615, 2023; *J. Med. Chem.* 66, 3284-3300, 2023). In addition, we conducted two-way ANOVA for the groups that were treated with either of the three compounds at two concentrations. Please note that the vehicle-treated group cannot be included in this analysis. The results are presented below.

	0.1 $\mu\text{mol T}_3$	1 $\mu\text{mol T}_3$	0.1 $\mu\text{mol GC-1}$	1 $\mu\text{mol GC-1}$	0.1 $\mu\text{mol ZTA-261}$	1 $\mu\text{mol ZTA-261}$	Effect of compound	Effect of Dose	Interaction
Body weight (g)	39.3 \pm 1.1	34.7 \pm 1.0	33.2 \pm 0.9	33.3 \pm 0.6	41.5 \pm 1.3	35.5 \pm 0.8	$p < 0.0001$	$p < 0.0001$	$p = 0.0068$
Epididymal adipose weight (mg)	1762.6 \pm 148.0	1015.0 \pm 36.4	1485.7 \pm 103.3	1093.5 \pm 50.3	2649.6 \pm 146.3	1679.3 \pm 109.0	$p < 0.0001$	$p < 0.0001$	$p = 0.0323$
Liver triglyceride (mg/g liver)	66.54 \pm 5.46	43.04 \pm 3.90	36.11 \pm 5.30	43.75 \pm 4.29	39.81 \pm 8.66	30.01 \pm 4.48	$p = 0.0022$	$p = 0.0657$	$p = 0.0258$
Liver cholesterol (mg/g liver)	1.83 \pm 0.07	1.96 \pm 0.13	1.99 \pm 0.09	2.10 \pm 0.13	1.69 \pm 0.11	1.95 \pm 0.14	$p = 0.1375$	$p = 0.0749$	$p = 0.7966$
Liver total lipid (mg/g liver)	30.0 \pm 1.8	22.1 \pm 0.7	19.1 \pm 1.3	20.7 \pm 1.1	18.8 \pm 2.1	17.1 \pm 1.2	$p < 0.0001$	$p = 0.0303$	$p = 0.0067$
Serum triglyceride (mg/dL)	95.26 \pm 6.13	130.5 \pm 8.92	62.68 \pm 7.05	93.92 \pm 6.17	84.35 \pm 4.25	75.25 \pm 4.44	$p < 0.0001$	$p = 0.0005$	$p = 0.0014$
Serum cholesterol (mg/dL)	129.83 \pm 7.30	91.58 \pm 2.82	103.23 \pm 3.91	106.61 \pm 3.48	172.98 \pm 5.35	124.09 \pm 6.40	$p < 0.0001$	$p < 0.0001$	$p < 0.0001$
Oil Red O-positive area (%)	3.46 \pm 0.54	2.17 \pm 0.44	1.40 \pm 0.57	1.85 \pm 0.49	2.06 \pm 0.70	0.19 \pm 0.09	$p = 0.0050$	$p = 0.0337$	$p = 0.0668$
Serum ALT (mU/mL)	11.35 \pm 1.36	18.90 \pm 2.78	13.15 \pm 1.28	23.62 \pm 4.61	7.67 \pm 0.68	10.80 \pm 0.93	$p = 0.0012$	$p = 0.0006$	$p = 0.3065$
Heart weight (mg)	192.6 \pm 4.9	236.1 \pm 4.5	130.6 \pm 3.9	173.5 \pm 5.1	135.1 \pm 3.1	144.1 \pm 3.1	$p < 0.0001$	$p < 0.0001$	$p < 0.0001$
Cardiomyocyte width (μm)	16.2 \pm 0.3	17.5 \pm 0.2	13.0 \pm 0.3	14.6 \pm 0.3	12.2 \pm 0.2	12.3 \pm 0.2	$p < 0.0001$	$p < 0.0001$	$p = 0.0127$

Data are shown as mean \pm SEM (n=8-10)

Responses to Reviewer 5

We are grateful to Reviewer 5 for the critical comments and useful suggestions that helped us improve our manuscript. As indicated in the following responses, we have considered all the comments and suggestions in the revised version of our manuscript.

REMARKS TO THE AUTHORS

The manuscript deals with the design and synthesis of a new agonist of THR β , endowed of selectivity against THR β . The new compound synthesized was evaluated as potential therapeutic agent for hyperlipidemia and obesity.

The new molecule showed to reduce hepatic lipid droplets in HFD induced obesity model mice. The authors assessed also the cardio and hepatotoxicity comparing the results with GC-1, a THR β selective agonist which terminated Phase I clinical trials because of cardiotoxicity.

The manuscript is in a hot topic field and numerous researches are currently focused on the search for new THR beta agonists to treat NASH, NAFLD and HCC. Although it deserves publication few revisions are needed before to accept it.

1. Title should be changed, and written in a more catchy way. Just to suggest one example: Design and synthesis of ZTA-261: a selective beta-subtype agonist of thyroid hormone receptor with low toxicity

RESPONSE

Thank you for suggesting a title that effectively summarizes this article. We would like to use it.

2. Abstract: I suggest to write a concise and clear abstract that resumes the results of the study. In such way, the abstract is mainly a brief introduction on the THR's biological activity.
3. Add in the abstract also the analysis of gene expression

RESPONSE

As per the suggestion of the reviewer, we have rewritten the abstract.

4. References should be updated. Several papers have been recently published describing novel THRbeta agonists (Li, Qiu, et al. Discovery of a highly selective and H435r-sensitive thyroid hormone receptor β agonist. *Journal of Medicinal Chemistry*, 2022, 65.10: 7193-7211; *Eur J Med Chem*. 2020 Feb 15;188:112006. doi: 10.1016/j.ejmech.2019.112006.) More importantly on January 2023 Resmetirom has been approved by FDA as the first THRbeta agonist to treat NASH. The authors must include these references and update the introduction with current state-of-the-art in this field.

RESPONSE

As per the reviewer's suggestion, we cited the papers and updated the current state of the development of THR agonists in the discussion section.

To assess whether the effect on lipid metabolism is mediated by activation of THR, a qRT-PCR analysis of THR-target genes, such as *Dio1*, *Thrsp*, and *Me1* should be performed (ref.: <https://doi.org/10.3390/ijms222313105>)

RESPONSE

We conducted further RT-qPCR analysis on three genes regulated by THR: *Dio1*, *Me1*, and *Thrsp*. The expression of these genes was upregulated by ZTA-261, as well as by T3 and GC-1. These findings suggest that ZTA-261 regulates lipid metabolism through the thyroid hormone signaling pathway.

5. Please check ref 20. The citation seems to be wrong. The article cited describes the use of fluorinated sulfone in cross-coupling reaction, but the authors cite this paper to justify the replacement of methyl groups of GC-1 with CF₃ moiety.

RESPONSE

We corrected the numbering of references. Thank you so much for a comment from a reviewer.

6. Synthetic procedures: Please add in Reagent and condition (figure 1c) the reaction yields. In the supporting info the purity of the final compounds should be assessed by HPLC or elemental analysis and data should be added in the experimental part.

RESPONSE

Thank you so much for a comment. We revised Figure 1c to add yields at each step. And we checked the purity of the final compounds by R-HPLC analysis in SI.

7. What is the difference between table 1 and table S1 (IC₅₀ values of compounds)? Please explain and correct, appropriately

RESPONSE

The results shown in Table S1 and Figure S1 were obtained through a ligand binding assay using 9XGal4UAS-luc2P reporter cell lines, conducted as the initial screening of compounds. In this assay, the ligand binding domain of THR_s was used, which results in slightly different results from the T₃ displacement assay shown in Fig. 2 and Table 1. From a quantitative standpoint, the in vitro T₃ displacement assay is considered more reliable than the cell-based assay as it uses full-length THR_s.

8. In the discussion section, the authors justify the results, hypothesizing an improved bioavailability of ZTA-261 when com and an improved distribution in the liver compared with plasma. These speculations should be confirmed by experimental data. I suggest to discuss only

results emerging from this study and clearly declare that further study should be performed in order to attribute these results to a different PK profile.

RESPONSE

We completely agree that demonstrating the pharmacokinetics profile with wet experimental data is very important to prove that ZTA-261 is a better THR-beta selective ligand. However, the current paper describes a study that focuses on and evaluates the biological responses of ZTA-261, such as receptor selectivity, fat accumulation in mice, and effects on bone weight. Thank you so much for pointing this out. We are hoping to report the more detailed mechanism of the compound, including PK and PD profiles and exploring the docking mode, along with further studies toward the development of THR-beta selective drug in the near future.

Reviewers' comments:

Reviewer #2 (Remarks to the Author):

Nambo M. and colleagues uncovered a new T3 analog, ZTA-261, with a higher selectivity for THR β over THR α when compared to GC-1. Although ZTA-261 exhibits lipid-lowering properties similar to GC-1, this discovery could represent a novel therapy for treating patients affected by hypothyroidism, particularly in individuals with heart complications, and for reducing lipid levels without causing cardiac thyrotoxicosis.

Reviewer #3 (Remarks to the Author):

Main comments

1. Please add the group size (number of mice used) in the legend and materials and methods.
2. In the data, authors compared control with T3 and its agonist using One-Way ANOVA. Authors may include multiple comparisons to show whether there is a difference resulting from the treatment using GC-1 and ZTA-261 in serum cholesterol, TG, and other testing parameters. Usually, the one-way Anova analysis includes multiple comparison automatically (depending on which statistical software you are using). If it is not included, the comparison between GC-1 and ZTA-261 should be compared separately.
3. The THR β -binding kinetics showed that ZTA-261 has better selectivity for THR β than GC-1, though ZTA-261 is not as effective as GC-1 in reducing epididymal fat (Figure 3B). Authors may comment on that, e.g., white fat is considered more THR α action than THR β action, which may explain why ZTA-261 is less effective in reducing white fat.
4. Although liver histology with Oil Red O staining (Figure 5) showed ZTA-261 is more effective in reducing lipid, the liver TG level (Figure 4) showed there is no significant difference in TG level between ZTA-261 and GC-1 treatment (Figure 4E). Authors explain the inconsistencies. It is possible that histology showed one sample, which may not be as accurate as the TG assay using multiple samples.
5. Figure 6F: Please make sure you have a statistical comparison between ZTA-261 and GC-1. SREBP-1 controls de novo fatty acid synthesis in the liver. If there is significantly higher SREBP-1 in ZTA-261 treated compared to GC-1, then it should be mentioned.
Line 332, "Srfebp1c, activating the synthesis of triglyceride, is reduced." Please indicate what you compare with.
6. High-level expression of Pnpla2 reflects low TG in the liver. The authors should add the comparison of GC-1 with ZTA-261. If Pnpla2 is significantly higher than GC-1, then it is in favor of histology data. However, the authors must explain why there is no difference in TG level in the liver; perhaps the authors did not do multiple comparisons in the statistics.
7. Figure 7C it is not necessary. Hyperthyroidism-induced tachycardia is mainly due to ADR β 1 and the muscarinic acetylcholine receptor M2. The cardiomyocyte width does not reflect cardiomyocyte function, such as contractibility.
In my opinion, this figure does not add value to the study. It is fine without figure 7C. If authors

would like to check the cardiac effects of the ZTA-261, a simple EKG would do the job after injection in mice with a relatively high dose of agonist.

Reviewer #4 (Remarks to the Author):

The authors have addressed my concerns satisfyingly with the exception for the extra qpcrs, as they did not store the tissues.

Reviewer #5 (Remarks to the Author):

As already said, the manuscript is in a hot topic field and numerous researches are currently focused on the search for new THR beta agonists to treat NASH, NAFLD and HCC. The study has been performed following an appropriate methodology and results are interesting. The revised manuscript reports the main replies to Reviewers'suggestions and it deserves the publication

Responses to Reviewer 1

We are grateful to Reviewer 1 for the critical comments and useful suggestions, which have helped us improve our manuscript. As indicated in the following responses, we have addressed all of the comments and suggestions in the revised manuscript.

REMARKS TO THE AUTHOR

Authors reported a new TRb-selective analog, ZTA-261. Authors showed that ZTA-261 is a more TRB-selective agonist than GC-1. ZTA-261 has dramatically lesser selectivity for TRa than GC-1, suggesting a lesser effect on the heart, muscle, and bone when used in the treatment. ZTA-261 is more effective in reducing lipid droplets in HF diet-induced fatty livers compared to GC-1. Authors showed limited ZTA-261 toxic analysis for bone, liver, and heart. ZTA-261 may be worth studying further for its utility. In this revised manuscript, the authors added new data and improved quality. There are some questions that remain, and modifications are needed.

1. BBB permeability test

Authors added this important data. The BBB permeability of ZTA-261 is ~50% lower than GC-1 and 10 times lower than caffeine, which means ZTA-261 will require a transporter to cross the BBB. Less permeability means less effects to brain and H.P.T. axis, which may be an advantage if you want it just for treating liver hyperlipidemia. Does ZTA-261 require transporter to get into the tissues? It is not known at this stage of the study. So far, GC-1 transporter has not been identified. Authors need to extend their data explanation.

RESPONSE

As pointed out by Reviewer 1, the permeability of ZTA-261 was lower than that of GC-1, which is beneficial for the treatment of hyperlipidemia without affecting the CNS. Unfortunately, transporter(s) for GC-1 and ZTA-261 have not been reported. To address this point, we have added following sentences to the revised manuscript (lines 291–293): “This result suggests that ZTA-261 is particularly useful for the treatment of liver hyperlipidemia, without affecting the CNS. Future studies are needed to identify transporters for GC-1 and ZTA-261”.

2. Concentrations of T3 and GC-1 and ZTA-261 used in animal experiments

In the Methods, authors stated that mice with T3 and its agonist at 0.1 and 1 $\mu\text{mol/kg/day}$ for 3

weeks. The doses used in animal experiments are high. Authors should provide rationale and/or references for high-dose treatment. In published data, the GC-1 concentration used in mouse injection is generally between 1.5 and 32 ug/kg/day. The T3 concentration used varies depending on the purpose, such as to render hyperthyroidism, deplete TSH or treat hypothyroidism; it ranges from 25–100 ug/100 bw/day for 3–8 days (usually). For depletion of TSH, a higher dose of T3 (250 ug/kg/day) for 6 days is sufficient.

For the initial study of ZTA-261, using a high dose may be necessary, but authors need to explain their rationale or cite the study that used it and why they used it.

After 3 weeks of treatment, the serum T3 level and/or TSH level should be presented in the manuscript. These are important data and should not be in the supplemental. Based on BBB permeability data, ZTA-261 is not expected to change serum TSH or T3.

With ZTA-261 (1 umol/kg/day) treatment for 3 weeks, liver lipid accumulation was significantly reduced without changing ALT, which is an important potential for the compound. However, the side effects on the liver from long-term treatment are not known. Therefore, further dosing and duration studies are needed, which authors should explain, point out, or discuss adequately.

RESPONSE

We determined the dosages based on a study by Johansson et al. ^[22] and have added this reference to the text (line 154). This study demonstrated that administration of T₃ and GC-1 at 97 nmol/kg/day efficiently reduced serum and liver cholesterol and triglyceride levels in mice. Therefore, we used a dose of 0.1 μmol/kg/day. In addition to this dosage, we selected a 10 times higher dose (i.e., 1 μmol/kg/day) to study potential side effects. As per the reviewer's suggestion, we have mentioned the importance of evaluating a wider range of doses and treatment durations in the revised manuscript as follows (lines 382–385): “In this study, we administered a higher dose of compounds for a longer duration than usual to investigate potential side effects caused by THR α . In future studies the dosage and duration should be carefully considered to further evaluate the effects of ZTA-261”.

As per the reviewer's suggestion, we have moved the data on serum T₃ levels from the Supplementary Information to the main manuscript (Table 3). Based on the BBB permeability data, the migration of GC-1 and ZTA-261 to the brain is limited. However, the BBB does not limit transport to the circumventricular organs, including the pituitary gland. Therefore, GC-1 and ZTA-261 may reduce serum T₃ levels via short-loop feedback on the pituitary gland. We expected that the decrease in endogenous serum T₃ would be compensated by exogenous GC-1 and ZTA-

261. We have summarized these points in the following revised sentences (lines 301–304): “The increase in the T₃ levels in the T₃-injected group may reflect exogenous T₃. It is possible that GC-1 and ZTA-261 inhibited endogenous T₃ synthesis and secretion by acting on the pituitary, which does not have a BBB, but not on the hypothalamus”.

3. Gene expression.

Please spill out the full name of each gene when it first appears.

Thrsp, Ldlr, and Cyp7a1 are well-known T₃-target genes. In the authors’ data, these genes are not stimulated by either T₃. If your experimental data is unexpectedly inconsistent with decades’ published data, you must find out the reason. Authors cannot simply say “reason unknown” without giving any explanation.

T₃ did not stimulate Ldlr, Cyp7a1 and Thrsp gene expression because of the extremely high concentration used in the mouse treatment. At excessively high concentrations of T₃, Ldlr mRNA is either reduced or not induced (please check out published data).

GC1 is similar to T₃ in stimulating Cyp7a1 and Thrsp gene expression. However, at excessively high concentrations, GC-1 lost its stimulative properties. There are published papers with dose-response studies available on the PubMed (Endocrinology 2012, PMID: [22067320](https://pubmed.ncbi.nlm.nih.gov/22067320/); PNAS 2005 <https://doi.org/10.1073/pnas.0504379102...>)

RESPONSE

Based on the reviewer’s suggestion, we have included the full names of genes at their first mention (lines 195–196, 202–205).

In this study, treatment with GC-1 and ZTA-261 increased the expression of *Thrsp* significantly. Additionally, T₃ treatment tended to increase *Thrsp* expression.; however, the change was not significant. Therefore, the results for *Thrsp* expression were consistent with those of previous studies.

It has been reported that TH reduces serum cholesterol levels by stimulating the expression of *Ldlr*. However, experiments using *Ldlr*^{-/-} mice have shown that TH can decrease serum cholesterol levels independently of *Ldlr*^[36, 39]; additionally, GC-1 treatment reduced *Ldlr* expression^[39]. Therefore, treatment with TH and its analogs may not necessarily increase *Ldlr* expression.

In the study by Johansson et al^[22], *Cyp7a1* was inhibited by 100 nmol/kg/day. Although *Cyp7a1* is a TH target gene, it is possible that the stimulation of *Cyp7a1* expression is dose-dependent. Therefore, the high dose used in this study may have inhibited the expression of *Cyp7a1*.

We have discussed the points mentioned above in the revised manuscript as follows (lines 325–336). “Previous studies have shown that GC-1 reduces serum cholesterol levels by inducing *Cyp7a1*^[36, 37]. However, another study has shown that these compounds have dose-dependent effects on expression levels of *Cyp7a1* in mice^[22]. The greatest increase in *Cyp7a1* was observed at a dose of 20 nmol/kg/day for both T₃ and GC-1, with a gradual decrease as the dose increased^[22]. In this study, *Cyp7a1* expression was not affected by T₃, GC-1 or ZTA-261 (Figure 6, D and E), which may be explained by the high doses (i.e., 0.1 and 1 mmol/kg/day).

TH is believed to reduce serum cholesterol levels by stimulating *Ldlr*^[38]. However, experiments using *Ldlr*^{-/-} mice have shown that TH can decrease serum cholesterol levels independently of *Ldlr*^[22, 36, 37, 39]. Furthermore, GC-1 treatment not always induced *Ldlr* expression^[36]. These findings indicate that treatment with TH and its analogs does not necessarily increase *Ldlr* expression”.

Minor: remove unnecessary phrases.

- For example, “saline solution of the various compound was used ...”

Not necessary to mention it. In fact, you are not allowed to use other type of solution for injection in animal experiment.

If an experimental method is already described in the Materials & Methods, there is no need to repeat the details in the Results. Condense the Line 157-162. Authors check through the manuscript and do the necessary changes.

- **Authors check your manuscript carefully for unnecessary phrases.**

I’m only pointing out a few examples below:

“with these consideration in mind”

“with these basic structure elements in mind”

“with this new hormone derivates in hand”,

“Lipid droplets are organelles for the storage of neutral lipids such as triglycerides and cholesterol esters.” ---- a common knowledge

Line 200-201, the sentence repeats the subtitle, consider either remove or change.

RESPONSE

We reviewed the manuscript based on the detailed comments and made the following changes:

- In the "Effects of TH analogs on body weight and accumulation of visceral fat" section, we have removed unnecessary explanations and sentences that overlap with those in the Materials and Methods (lines 152–166).
- We have removed the sentence "Lipid droplets are organelles for the storage of neutral lipids such as triglycerides and cholesterol esters." from the "Histological evaluation of lipid accumulation in the liver" section (lines 189–190 of the previous version).
- We have changed the subsection heading "Expression analysis of the THR-regulated genes and genes involved in the lipid metabolism in the liver" to "Gene expression analysis in the liver" to minimize redundancy (line 181).
- We have revised the following phrases (lines 111–133)
Original: "With these considerations in mind"
Revised: "Therefore"
Original: "With these structures in mind":
Revised: removed
Original: "With these new hormone derivatives in hand, their affinity and selectivity for THR α and β were tentatively evaluated by a cell-based assay."
Revised: "The affinity and selectivity of these new hormone derivatives for THR α and β were tentatively evaluated by a cell-based assay"

Responses to Reviewer 3

We are grateful to Reviewer 3 for the critical comments and useful suggestions, which have helped us improve our manuscript. As indicated in the following responses, we have addressed all of the comments and suggestions in the revised manuscript.

Main comments

1. Please add the group size (number of mice used) in the legend and materials and methods.

RESPONSE

As per the reviewer's suggestion, we have added the following sentence "Each group consisted of 10 animals, except the ND (n=8) and HFD with saline (Vehicle, n=9) groups." (Materials and Methods, lines 440–441). We have also revised the description of the group sizes in the legend of Figure 3 as follows, "Data are shown as mean \pm SEM (n=8-10)" (lines 770–771).

2. In the data, authors compared control with T3 and its agonist using One-Way ANOVA. Authors may include multiple comparisons to show whether there is a difference resulting from the treatment using GC-1 and ZTA-261 in serum cholesterol, TG, and other testing parameters. Usually, the one-way Anova analysis includes multiple comparison automatically (depending on which statistical software you are using). If it is not included, the comparison between GC-1 and ZTA-261 should be compared separately.

RESPONSE

We have re-analyzed the data presented in Figures 3–6 using GraphPad Prism 8. For pairwise comparisons of mean values, we used one-way ANOVA followed by Tukey's multiple comparison tests. Alternatively, to compare all possible pairs of mean ranks, we used the Kruskal-Wallis test followed by Dunn's multiple comparison tests. As shown in Figures 5 and 6, values in the T₃, GC-1, and ZTA-261-treated groups differed significantly from those in the control group, and there were no significant differences among the T₃, GC-1, and ZTA-261-treated groups. Therefore, we have deleted the discussions of the comparison between the GC-1- and ZTA-261-treated groups (See below).

3. The THR_b-binding kinetics showed that ZTA-261 has better selectivity for THR_b than GC-1, though ZTA-261 is not as effective as GC-1 in reducing epididymal fat (Figure 3B). Authors may comment on that, e.g., white fat is considered more THR_α action than THR_b action, which may explain why ZTA-261 is less effective in reducing white fat.

RESPONSE

As suggested by Reviewer, we have added a sentence in the Discussion as follows “Since white fat is considered to have more THR_α activity than THR_β activity, the reduction in epididymal fat by GC-1 may also reflect the degree of THR_β selectivity.”(lines 368 –370).

4. Although liver histology with Oil Red O staining (Figure 5) showed ZTA-261 is more effective in reducing lipid, the liver TG level (Figure 4) showed there is no significant difference in TG level between ZTA-261 and GC-1 treatment (Figure 4E). Authors explain the inconsistencies.

It is possible that histology showed one sample, which may not be as accurate as the TG assay

using multiple samples.

RESPONSE

Representative images of Oil Red-O-stained liver tissues from each animal (n=8–10) are shown in Figure S3. The distribution of signal intensities was consistent with the results summarized in Figure 5B.

Lipid droplets in the liver increase not only in NASH but also in drug-induced liver injury^[40]. Animals treated with GC-1 at 1 $\mu\text{mol/kg/day}$ showed significantly higher serum ALT levels than those in the ZTA-261 treated group. This suggests that the observed lipid droplets may have been caused, in part, by GC-1-induced liver injury.

We have discussed these points in the revised manuscript as follows: “ZTA-261 was more effective in reducing fat droplets in the liver. The administration of ZTA-261 at 1 $\mu\text{mol/kg BW/day}$ reduced fat droplets to a level comparable to that in animals fed a normal diet. In contrast, liver TG levels were similar in the GC-1- and ZTA-261-treated groups. This discrepancy may be explained by the fact that the number of lipid droplets can also be increased by drug-induced liver injury^[40]. In this study, serum ALT levels were higher in GC-1-treated groups than in ZTA-261-treated groups. The droplets observed in the GC-1-treated group could be explained, in part, by the side effects of GC-1 on the liver.” (Lines 345–352).

5. Figure 6F: Please make sure you have a statistical comparison between ZTA-261 and GC-1. SREBP-1 controls de novo fatty acid synthesis in the liver. If there is significantly higher SREBP-1 in ZTA-261 treated compared to GC-1, then it should be mentioned.

Line 332, "Srfebp1c, activating the synthesis of triglyceride, is reduced." Please indicate what you compare with.

6. High-level expression of Pnpla2 reflects low TG in the liver. The authors should add the comparison of GC-1 with ZTA-261. If Pnpla2 is significantly higher than GC-1, then it is in favor of histology data. However, the authors must explain why there is no difference in TG level in the liver; perhaps the authors did not do multiple comparisons in the statistics.

RESPONSE

We used Kruskal-Wallis tests followed by Dunn’s multiple comparison tests for pairwise comparisons in Figure 6F. The levels of *Srebp1c* expression were significantly lower in the groups treated with T₃ and GC-1 than in the control group; however, there was no significant difference

between the ZTA-261-treated and control groups. Among the T₃-, GC-1-, and ZTA-261-treated groups, no significant difference was observed in the expression of *Srebp1c*. We also re-analyzed the data used to generate Figure 6G; we observed a significant difference in the expression of *Pnpla2* between the ZTA-261-treated and control groups; however, we did not observe significant differences among the groups treated with T₃, GC-1 and ZTA-261. We have revised the description of these analyses as follows: “The expression levels of *Srebp1c*, which encodes a transcription factor that activates the synthesis of fatty acids and triglycerides, were significantly lower in the T₃, and GC-1 groups than in the vehicle group. Although not statistically significant, the ZTA-261-treated group showed decreased *Srebp1c* expression (Figure 6F). Levels of *patatin-like phosphodomain-containing 2 (Pnpla2)*, also known as *adipose triglyceride lipase (ATGL)*, which encodes an enzyme that catalyzes the hydrolysis of triglycerides^[25], were significantly higher in the ZTA-261-treated group than in the vehicle group (Figure 6D).” (lines 209–216), and “Treatment with T₃ and GC-1 significantly decreased the expression of *Srebp1c*, activating the synthesis of triglycerides, and treatment with ZTA-261 significantly upregulated the expression of *Pnpla2*, regulating the degradation of triglycerides (Figure 6, F and G) “ (lines 339 to 342). These results indicate that the difference in the liver histology and TG levels in the liver cannot be attributed to differences in the expression of these genes. Therefore, we have removed the following sentences: “The increase in the expression level of *Pnpla2* was more prominent in the ZTA-261 treated groups than in the T₃- and GC-1-treated groups (Figure 6G). This could explain why ZTA-261 reduced the liver fat droplets more effectively than T₃ and GC-1.” (lines 331–334 in the previous version)

As mentioned earlier, the lipid droplets observed in the GC-1-treated groups may have been caused, in part, by GC-1-induced liver injury.

7. Figure 7C it is not necessary. Hyperthyroidism-induced tachycardia is mainly due to ADRb1 and the muscarinic acetylcholine receptor M2. The cardiomyocyte width does not reflect cardiomyocyte function, such as contractibility.

In my opinion, this figure does not add value to the study. It is fine without figure 7C. If authors would like to check the cardiac effects of the ZTA-261, a simple EKG would do the job after injection in mice with a relatively high dose of agonist.

RESPONSE

Increases in contractility and heart rate caused by hyperthyroidism may lead to cardiomyocyte hypertrophy (Reviewed in *Endocr. Connect.* 9, R59, 2020). Therefore, we considered the cardiomyocyte width as a potential indicator of cardiac dysfunction resulting from excess TH. We would prefer to keep the diagram in its original form; however, we will follow the editor's suggestion.

REVIEWERS' COMMENTS:

Reviewer #3 (Remarks to the Author):

Review comments

Nambo M. et al. reported a new T3 analog ZTA-261. TH influence many biological processes especially in lipid metabolism. TH treatment has limitations due to thyrotoxicosis. Several T3 analogues had gone clinical trial and terminated at phase 3 trials due to side effects on cartilage in animals with long-term treatment. However, T3 analogues lipid lowering properties are attractive for NAFLD. GC-1 is one of the well-studied T3 analogues for its lipid lowering properties. In the second revision of the manuscript, the authors carefully revised statements and further clarified the data. Selectivity is a crucial factor for T3 analogues. The study showed the ZTA-261 has good selectivity for TRb compared to GC-1. Regarding the lipid lowering properties, it is comparable to GC-1 in reducing serum cholesterol and TG. In mice fed with high-fat diet, ZTA-261 showed better results than GC-1. ZTA0-261 also showed exceptionally low ability to cross the BBB, meaning the analogue has little effects on T3-mediated function in the brain. These features indicate ZTA-261 has the potential for translational utility for NAFLD. Authors has completed the initial study of this compound for its selectivity, lipid lowering properties in whole animals with HFD and liver, effects on cardiomyocytes, and BBB crossing. The article deserves acceptance for publication.